# *From Fragments to Facts:* A Curriculum-Driven DPO Approach for Generating Hindi News Veracity Explanations

## Abstract

In an era of rampant misinformation, generating reliable news explanations is vital, especially for under-represented languages like Hindi. Lacking robust automated tools, Hindi faces challenges in scaling misinformation detection. To bridge this gap, we propose a novel framework integrating Direct Preference Optimization (DPO) with curriculum learning to align machine-generated explanations with human reasoning. Fact-checked explanations from credible sources serve as preferred responses, while LLM outputs highlight system limitations and serve as non-preferred responses. To refine task-specific alignment, we introduce two key parameters—*Actuality* and *Finesse*—into the DPO loss function, enhancing explanation quality and consistency. Experiments with LLMs (Mistral, Llama, Gemma) and PLMs (mBART, mT5) confirm the framework's effectiveness in generating coherent, contextually relevant explanations. This scalable approach combats misinformation and extends automated explanation generation to low-resource languages.

## 1 Introduction

The rise of fake news, fuelled by its low production cost and widespread digital dissemination, poses a significant threat to the integrity of journalism. As Toomas Hendrik Ilves aptly states, "*Fake news is cheap to produce. Genuine journalism is expensive.*" This disparity is evident in the societal disruptions caused by misinformation, particularly during crises like the COVID-19 pandemic, where unverified claims about treatments and preventive measures fuelled panic and confusion (Barua et al., 2020; Roozenbeek et al., 2020). Misinformation also exacerbates political polarization, creating deep societal divides (Cantarella et al., 2023; Bovet & Makse, 2019). Fact-checking platforms, while essential, face substantial challenges in scaling their efforts, especially in languages like Hindi. Despite over 600 million Hindi speakers[1], automated tools for generating credible, human-like explanations in Hindi remain underdeveloped. Addressing this gap is crucial to supporting fact-checking initiatives and reducing the impact of fake news on society.

**Overview:** To address these challenges, this paper introduces the ***DeFactoX*** framework, which evaluates the veracity of Hindi news articles—while remaining adaptable to other languages—and generates coherent, factually grounded explanations supporting its prediction. By analyzing the content, *DeFactoX* determines whether a news piece is credible or misleading and provides a well-reasoned justification, enhancing transparency and trust in automated misinformation detection.

We create a synthetic Hindi preference dataset by using fact-checked *explanations written by humans as preferred responses* and *machine-generated outputs from various LLMs as rejected responses*. To enhance the generation of high-quality explanations, we improve the Direct Preference Optimization (DPO) approach (Rafailov et al., 2024) by integrating fact scores, variance-based reduction parameters, and curriculum learning. These enhancements align generated explanations with human reasoning, ensuring reliability and accuracy. Leveraging curriculum learning's data augmentation, our scalable framework adapts to low-resource languages, aiding global misinformation detection and automated explanation generation.

---

[1]https://www.britannica.com/topic/languages-by-total-number-of-speakers-2228881

**Research Gap:** While NLP has made significant strides, most automated explanation generation systems focus on high-resource languages like English and Chinese (Wang et al., 2020; Zhang et al., 2020; Xu et al., 2024; Hsu et al., 2023) , leaving Hindi largely underserved.

Pre-trained LLMs, trained on generalized datasets, struggle to assess the veracity of Hindi news and generate contextually relevant, factually grounded explanations for the veracity predicted. Moreover, fact-checking in Hindi remains predominantly manual, lacking scalable automated solutions. Given Hindi's vast speaker base and the increasing spread of misinformation, it is crucial to develop robust, scalable methods for **automated veracity prediction and explanation generation.** While *our framework is language-agnostic*, we focus on Hindi to address this pressing need and demonstrate its effectiveness in a low-resource setting.

**Research Questions:** This research aims to address the following questions:

- **RQ-1:** How can automated systems reliably assess the veracity of Hindi news and generate human-like explanations that are coherent, contextually relevant, and factually accurate while explicitly justifying the model's veracity predictions?

- **RQ-2:** Can Direct Preference Optimization (DPO) (Rafailov et al., 2024), augmented with FactScore (Min et al., 2023) and variance-based parameters, effectively align machine-generated explanations with human standards in resource-constrained languages like Hindi, ensuring both factual accuracy and interpretability?

- **RQ-3:** How can curriculum learning (Pattnaik et al., 2024) be integrated with DPO to refine veracity prediction in under-represented languages, and what scalable methodologies can extend misinformation detection and explanation generation to other low-resource languages?

**Research Motivation:** The growing spread of misinformation in underrepresented languages like Hindi underscores the urgent need for scalable, automated systems that can assess veracity and generate reliable, human-like explanations. Unlike high-resource languages, Hindi suffers from a lack of robust fact-checking tools, and existing LLMs often fall short of maintaining coherence, factual accuracy, and human alignment for veracity explanations.

This research aims to bridge these gaps by leveraging Direct Preference Optimization (DPO) (Rafailov et al., 2024), curriculum learning (Pattnaik et al., 2024), FactScore (Min et al., 2023), and variance-based reduction techniques to refine veracity explanation generation, ensuring both accuracy and scalability in combating misinformation.

**Contributions** The key contributions of our research are as follows:

- We introduce a *synthetic, ranking-based Hindi preference dataset*, where human-written fact-checked veracity explanations serve as the highest-ranked responses, while machine-generated outputs from various LLMs are evaluated and ranked based on a *weighted scoring mechanism* incorporating BERTSCORE (Zhang et al., 2019), ROUGE-L (Lin, 2004), and METEOR (Banerjee & Lavie, 2005). This ensures that while machine-generated explanations are treated as non-preferred responses, they are ranked based on quality rather than being uniformly categorized as poor.
- We introduce an enhanced DPO framework, ***DeFactoX***, inspired by the principle of clear, fact-based, human-written veracity explanations. Our framework integrates *FactScore, variance-based reduction parameters, and curriculum learning* to train models that not only predict veracity but also generate explanations that are coherent, contextually accurate, and aligned with human reasoning, thereby strengthening the model's veracity claims.
- Our language-agnostic framework lays the groundwork for automated veracity-driven explanation generation in low-resource languages. *Focusing on Hindi due to its underrepresentation in fact-checking*, we present the first framework of its kind, contributing to the global effort to combat misinformation with a scalable and adaptable solution.

# 2    Related Works

We explore the growth of misinformation detection through veracity explanation alongside advancements in Preference Optimization techniques.

**Automated Misinformation Detection and Explanation Generation:** Recent studies have advanced misinformation detection and explanation generation. Joshi et al. (2023) integrated Domain Adversarial Neural Networks (DANN) with LIME to enhance COVID-19 misinformation detection. Chi & Liao (2022) proposed QA-AXDS, a scalable, interpretable fake news detection system using dialogue trees. Yao et al. (2023) introduced MOCHEG, a multimodal fact-checking benchmark incorporating textual and visual evidence. Zhou et al. (2023) examined AI-generated misinformation, highlighting linguistic nuances and proposing updated detection guidelines. Gong et al. (2024) emphasized socio-contextual cues in "social explanation" to combat misinformation. Russo et al. (2023) showed that extractive steps improve abstractive summarization for claim verification. Bilal et al. (2024) developed a GNN-based rumour verification model leveraging opinion-guided summaries. Yue et al. (2024) introduced RARG, combining evidence retrieval with RLHF-tuned LLMs, excelling in COVID-19 misinformation detection. The study by Yang et al. (2022) employed a QA-based framework with attention-driven comparisons for interpretable fact-checking. For a comprehensive review, readers are pointed to the work by Kotonya & Toni (2020). ***Our Novelty:*** *While previous works primarily target high-resource languages like English and Chinese, our focus is on Hindi, an under-represented language.*

**Applications and Advancements in Preference Optimization and Curriculum Learning:** Recent advancements in preference optimization and curriculum learning have enhanced model performance across domains. Pattnaik et al. (2024) introduced Curry-DPO, a curriculum learning-based enhancement of DPO, achieving up to 7.5% improvement across datasets. Chen et al. (2024a) proposed a multi-stage curriculum framework optimizing humour and structure preferences in LLMs. Yin et al. Yin et al. (2024) developed Self-Augmented Preference Optimization (SAPO), surpassing DPO and SPIN across multiple benchmarks. Morimura et al. (2024) introduced filtered DPO (fDPO), refining datasets for better training efficiency. Wang et al. (2024) proposed Balanced Preference Optimization (BPO), enhancing knowledge depth while maintaining efficiency. Zeng et al. (2024) presented Token-DPO, improving alignment and diversity in LLMs through token-level fine-tuning. Wallace et al. (2024) introduced Diffusion-DPO, aligning diffusion models with human preferences for superior visual appeal. Chen et al. (2024b) proposed Softmax-DPO to enhance recommender systems via user preference modelling. Lai et al. (2024) introduced Step-DPO, improving mathematical reasoning in LLMs with minimal data. For a comprehensive review, readers are referred to the study by Xiao et al. (2024). ***Our Novelty:*** *While DPO has been applied across various domains, our approach is specifically tailored to generating veracity claims and explanations for Hindi news. The research gap in this area highlights challenges faced by under-represented languages like Hindi, including data scarcity and the limitations of LLMs trained on multilingual datasets. Our study offers a scalable and effective solution to address these challenges.* **Remark:** A detailed section on the theoretical foundations of DPO (Rafailov et al., 2024) and Curry-DPO (Pattnaik et al., 2024) is provided in the appendix (section A.5 & A.6) for interested readers.

# 3    Preference Dataset Creation

To construct our synthetic preference dataset, as shown in Figure 1, we followed a systematic multi-step approach to ensure data quality, uniformity, and relevance to the task. Below, we outline the process in a structured manner.

## 3.1    Dataset Selection and Sampling

Multiple veracity claim misinformation detection datasets Bhardwaj et al. (2020); Kumar & Singh (2022); Sharma & Arya (2024); Bansal et al. (2024); Sharma & Garg (2021); Badam et al. (2022); Sharma & Arya (2023) provide Hindi news articles sourced from fact-checking websites, ensuring authentic veracity labels. We

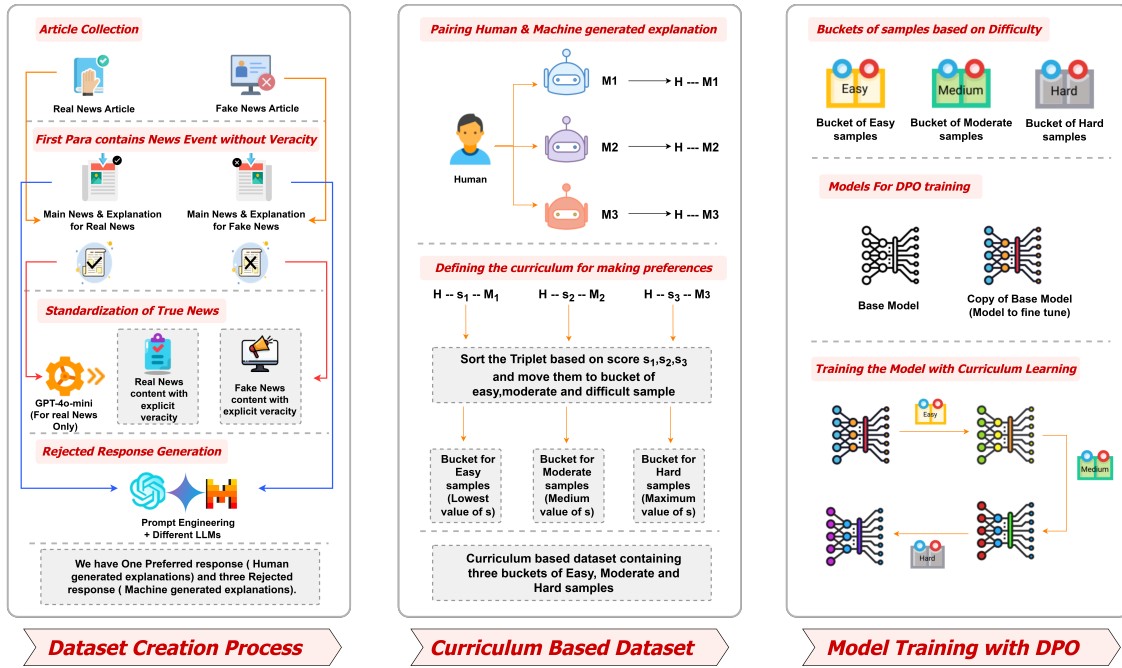

Figure 1: Overview of **DeFactoX** framework.

selected data instances from Sharma & Arya (2024), a comprehensive dataset featuring over 15,000 articles in the fake news category and 13,000+ in the real news category. This dataset was chosen for its extensive coverage of news, spanning from older to recent events, and its sourcing from fact-checking websites, which provide verified veracity labels. To maintain a balanced and manageable dataset, we extracted the **most recent** 3,000 articles from each class (fake and real news). The selection ensures a healthy mix of data samples without being overbearing in size.

## 3.2   Characteristics of the Selected Data

**(1)** Each article is sourced from reputable fact-checking websites that not only classify news as fake or real but also provide comprehensive, well-reasoned explanations justifying these classifications. These **explanations serve as ground truth references** for evaluating model-generated outputs.

**(2)** The first paragraph of every article **strictly contains only the core news content**, intentionally excluding any veracity indicators or reasoning. This neutral presentation ensures that readers, and more importantly, models—cannot determine whether the news is fake or real based solely on this segment.

This design choice is crucial for constructing the preference dataset, as these initial news passages act as inputs for models, which must then generate both veracity predictions (fake or real) and coherent supporting explanations.

## 3.3   Observations on Explanations

A key observation in our dataset is the **distinct difference in the writing styles of explanations for true and fake news**.

**(1) True news explanations:** Fact-checking sources often offer minimal explanations for true news, assuming that the facts and details presented are self-evident. These explanations are typically brief and rarely include explicit statements affirming the news' authenticity.

**Explanation of Why the News is Fake**

लेकिन सच ये है कि 'ये वायरल वीडियो दीया कुमारी का नहीं बल्कि गुजराती महिला निकिताबा राठौड़ का है। इस सच का खुलासा किया है आज तक चैनल ने, जिन्होंने गुजराती महिला निकिताबा राठौड़ से इस बारे में बात की और वायरल वीडियो का असली सच लोगों को बताया। निकिताबा राठौड़ ने 22 जनवरी यानी की अयोध्या में रामलला के प्राण-प्रतिष्ठा वाले दिन अहमदाबाद के नरोडा में आयोजित एक कार्यक्रम में हिस्सा लिया गया था, जहां लोगों ने प्रभु राम के अयोध्या वापसी पर 11 हजार दीए जलाकर दिवाली मनाई थी।

**English Translation**

But the truth is that this viral video is not of Diya Kumari but of a Gujarati woman Nikitaba Rathod. This truth has been revealed by Aaj Tak channel, who talked to Gujarati woman Nikitaba Rathod about this and told the real truth of the viral video to the people. Nikitaba Rathod had participated in a program organized in Naroda, Ahmedabad on 22 January, the day of Ram Lalla's Pran-Pratishtha in Ayodhya, where people celebrated Diwali by lighting 11 thousand diyas on the return of Lord Ram to Ayodhya.

Figure 2: Snippet of fake news explanation with explicit reasoning for its veracity.

**(2) Fake news explanations:** In contrast, explanations for fake news are far more detailed and explicit. Fact-checkers provide strong declarative statements, rejecting falsehoods and supplementing them with clear justifications, such as evidence-based counterarguments, source verification, and logical reasoning. This explicit mention of veracity is illustrated in Figure 2. To gain further insight into the nature of these explanations, readers are encouraged to refer to the original fact-checking sources, including OneIndia[2], Vishvas News[3], and Aaj Tak[4], which serve as the primary references for this dataset.

## 3.4 Standardizing True News Explanations

While explanations for fake news naturally include explicit reasoning and veracity statements, true news explanations often lack such clarity. Fact-checking sources assume that factual news is self-evident, leading to minimal justifications. This inconsistency makes it difficult for models to learn a uniform veracity-based explanation structure. To address this, we standardized true news explanations by ensuring they explicitly affirm their veracity while preserving factual integrity. This step aligns true news explanations with the structured reasoning seen in fake news explanations, thereby creating a more balanced training dataset. For implementation details and examples, including the standardization prompt, please refer to the appendix (Section A.2 & A.3).

## 3.5 Generating Rejected Responses

To construct the negative class (non-preferred or rejected responses), we used the **first paragraph of each article, presenting only the core news content, whether fake or true,** as input for three state-of-the-art LLMs: `gpt-4o-mini` Achiam et al. (2023), `Mistral-7B-v0.1` Jiang et al. (2023), and `gemini-1.5-flash`

---

[2]https://hindi.oneindia.com/fact-check/
[3]https://www.vishvasnews.com/
[4]https://www.aajtak.in/fact-check

Team et al. (2024). These models were selected due to their strong reasoning capabilities and proven performance in NLP tasks (Liu et al., 2024; Mathur et al., 2024; Siino, 2024a;b; Trott & Rivière, 2024; Sato et al., 2024).

**Hallucination Mitigation:** Numerous recent studies Rosset et al. (2023); Hu et al. (2023); Ghosh et al. (2024) have leveraged language models to generate synthetic preference datasets for reinforcement learning via human feedback (RLHF), inspiring our approach as well. A primary concern in generating synthetic explanations is the risk of hallucination—incorrect or fabricated details in the model's output. To mitigate hallucination, we designed a controlled prompt that explicitly restricted responses to the provided news content, minimizing the inclusion of extraneous information.

Additionally, we manually reviewed 1000 randomly sampled explanations from both classes, verifying factual accuracy. Key checks included identifying geographical inconsistencies (e.g., misrepresenting an Indian event as occurring in the USA), verifying temporal accuracy (ensuring dates and timelines matched the original article), detecting entity distortions (confirming that names of people, organizations, and locations were correctly retained), and assessing logical coherence (ensuring explanations aligned with the core facts without introducing contradictions). No significant errors were found across these dimensions, reinforcing the overall reliability of our dataset. For implementation details, including the full prompt used for generating rejected responses, readers are pointed to the appendix (section A.4).

## 3.6 Algorithmic Overview of Dataset Creation from Fact-Checked News Articles

---

**Algorithm 1** Dataset Creation from Fact-Checked News Articles

---

**Require:** Scraped news article $A$ containing main news $N$ and explanation $E$, Large Language Models $\{LLM_1, LLM_2, LLM_3\}$, Scoring function $S$
**Ensure:** Dataset $D$ with explanations categorized by quality
 1: **Step 1: Segregate Explanation and Main News**
 2: Extract main news $N$ and ground-truth explanation $E_{GT}$ from article $A$
 3: **Step 2: Generate LLM Explanations**
 4: **for** each model $LLM_i$ in $\{LLM_1, LLM_2, LLM_3\}$ **do**
 5:    Provide $N$ as input to $LLM_i$ and obtain predicted explanation $E_i$
 6: **end for**
 7: **Step 3: Compute Scores for Explanations**
 8: **for** each predicted explanation $E_i$ **do**
 9:    Compute score $S(E_i)$ using scoring function $S$
10: **end for**
11: **Step 4: Rank Explanations**
12: Sort explanations $\{E_1, E_2, E_3\}$ in ascending order based on $S(E_i)$
13: **Step 5: Bucketize Explanations**
14: Define three score-based categories:
   - **Low-quality bucket** $B_L \leftarrow$ Explanations with lowest scores
   - **Medium-quality bucket** $B_M \leftarrow$ Explanations with mid-range scores
   - **High-quality bucket** $B_H \leftarrow$ Explanations with highest scores
15: **Step 6: Construct Final Dataset**
16: Form dataset $D$ by concatenating explanations in the order:

$$D = B_L \cup B_M \cup B_H$$

17: **return** $D$

---

## 3.7 Final Preference Dataset Composition

The final synthetic preference dataset comprises:

- **Preferred outputs:** Explanations for fake news, sourced from fact-checking websites, and true news explanations, standardized using the prompt described in the appendix (Section A.2).
- **Non-preferred outputs:** Machine-generated explanations, based on the first paragraph of the news articles, produced using the prompt detailed in Section A.4 of the appendix, by three state-of-the-art LLMs: `gpt-4o-mini`, `Mistral-7B-v0.1`, and `gemini-1.5-flash`.

Each data sample contains one positive (preferred) explanation and three negative (rejected) explanations, ensuring a balanced dataset by maintaining a consistent 1:3 ratio. This structure provides equal exposure to both high-quality and suboptimal explanations, helping to improve distinction and generalization. Furthermore, examples of input news and their corresponding non-preferred outputs, are provided in the appendix (Section A.7).

# 4    Methodology of *DeFactoX*

Inspired by the work by Pattnaik et al. (2024), we incorporated curriculum learning to improve model robustness in distinguishing high-quality explanations. Specifically, we ranked the three non-preferred responses based on their alignment with ground truth explanations using a scoring function that integrates BERTSCORE (Zhang et al., 2019), ROUGE-L (Lin, 2004), and METEOR (Banerjee & Lavie, 2005). The training was structured progressively: the model first learned from rank-0 explanations (most aligned with ground truth), then rank-1, and finally rank-2 (least aligned). This curriculum learning strategy ensured a structured learning trajectory, starting with simpler cases before introducing more challenging distinctions. An overview of our framework is depicted in Figure 1. ***Metrics for Final Score Calculation to Sort Explanations:*** The final score (*fs*) for each LLM-generated explanation was computed as follows:

$$\text{fs} = \frac{\text{BERTSCORE} + 3 \times (\text{ROUGE-L} + \text{METEOR})}{4}$$

**Weighting and Balancing Metrics:** Although **BERTSCORE**, **ROUGE-L**, and **METEOR** are theoretically normalized between 0 and 1, their practical distributions vary considerably. **BERTSCORE** typically falls within **0.85–0.95**, reflecting high semantic similarity, whereas **ROUGE-L** and **METEOR** generally range from **0.2–0.4**, capturing structural coherence and lexical overlap. If all metrics were weighted equally, BERTSCORE would overwhelmingly influence the final score, skewing evaluations toward semantic alignment while undervaluing fluency and coherence.

To mitigate this imbalance, we empirically assigned a weight of 3 to ROUGE-L and METEOR, amplifying their impact and ensuring a more equitable contribution. The final division by 4 normalizes the weighted sum, preserving interpretability while preventing overemphasising any metric. This formulation effectively harmonizes complementary aspects of text quality, achieving a nuanced and fairer evaluation of generated summaries.

**Ranking and Curriculum Learning Strategy:** Explanations were ranked into three levels: **rank-0** (most aligned with ground truth), **rank-1** (moderately aligned), and **rank-2** (least aligned). This ranking guided our **curriculum learning strategy**, progressively exposing the model to harder samples. Training began with **rank-0 explanations** to establish a strong foundation, followed by **rank-1 samples** to introduce moderate variations. Finally, **rank-2 explanations**, which deviated most, were incorporated to refine the model's ability to discern subtle differences. This structured progression, akin to human learning, enhances robustness and generalizability.

**Enhancements with Direct Preference Optimization (DPO):** Applying Direct Preference Optimization (DPO) to news veracity prediction and explanation revealed key areas for improvement. To strengthen factual evaluation, we introduced two novel parameters: ***Actuality***, inspired by the Factscore metric (Min et al., 2023), and ***Finesse***, a variance-based measure. These additions refine the preference model by prioritizing factual accuracy and consistency.

**Actuality Prompt Details:** Inspired by Factscore (Min et al., 2023), the *Actuality* score ensures factual accuracy in news explanations without relying on external references. It leverages GPT-4o-mini's internal knowledge to assess correctness, filtering out misinformation while reinforcing fact-based alignment. Further detailed justification is presented below.

---

**Prompt for *Actuality* Score**

**Task:** *You will be given a news article. Follow these steps:*
  1. *Extract up to 15 of the most important and factually relevant sentences from the article.*
  2. *For each extracted sentence, assess its factual correctness:*
       - *Label each sentence as **1** if it is factually accurate.*
       - *Label it as **0** if it contains factual errors.*
  3. *Compute the **average** of all the labels (1s and 0s).*
**Output:** *Return only the factual consistency score as a single numerical value (e.g., 0.75). Do not include any additional explanations, calculations, or extracted sentences.* **Here is the news article: {article} Answer:**

---

## 4.1 Justification for *Actuality* Score

**Critical Role of Factual Accuracy:** The credibility of AI-generated explanations in news-related tasks is fundamentally dependent on factual correctness. Given the increasing prevalence of misinformation and the potential societal impact of inaccurate content, ensuring that explanations align with verifiable facts is paramount. However, existing evaluation methods often rely on surface-level lexical overlap or reference-based approaches, which fail to fully capture factual consistency. This necessitates a more robust, reference-free metric that can objectively assess factuality in an automated and scalable manner.

**Defining the *Actuality* Score:** To address this challenge, we introduce the Actuality score, a novel metric specifically designed to evaluate factual correctness in both preferred and rejected explanations. Unlike traditional evaluation methods such as Factscore Min et al. (2023), which require external references or access to predefined knowledge bases, our approach leverages GPT-4o-mini's extensive training data to assess factuality in a self-contained manner. This makes the Actuality score highly adaptable, allowing it to evaluate explanations without the limitations imposed by reference-dependent metrics.

**Why Reference-Free Evaluation?** While reference-based approaches offer certain advantages, they suffer from critical drawbacks when applied to news explanation generation: 1. Dependency on External Knowledge Sources: Many factual evaluation metrics require access to external databases, knowledge graphs, or human-curated references. This dependency introduces constraints on scalability and adaptability, particularly for real-time or emerging news topics where no authoritative references exist. 2. Inability to Handle Implicit Knowledge: News explanations often contain implicit contextual inferences that are not explicitly stated in reference materials. A reference-dependent approach may misjudge such explanations as incorrect simply due to the absence of direct overlap. 3. Bias in Reference Selection: Fact-based evaluation using predefined references can inadvertently introduce bias, as it assumes the selected sources are always accurate and comprehensive. In contrast, a reference-free approach allows for broader contextual understanding, reducing reliance on potentially outdated or incomplete references.

**Mitigating Misinformation and Enhancing Reliability:** A significant challenge in news explanation ranking is the presence of highly fluent yet factually misleading explanations. These explanations may appear convincing but contain subtle distortions, omissions, or unverifiable claims, leading to misinformation. The Actuality score systematically mitigates this issue by enforcing a fact-based assessment criterion: - If an explanation includes factual inconsistencies, it receives a lower Actuality score, even if it is well-structured and grammatically sound. - Explanations that demonstrate verifiable correctness and alignment with known facts are rewarded, ensuring that AI-generated content prioritizes factual integrity.

**Impact on Explanation Ranking and Model Training:** Even among high-ranked explanations, subtle factual inaccuracies can persist. By incorporating the Actuality score into our evaluation pipeline, we achieve three key improvements: 1. Stronger Fact-Based Alignment: Unlike traditional ranking methods that may favor well-written but speculative responses, our approach explicitly penalizes factual errors, ensuring

that only rigorously accurate explanations receive higher ranks. 2. Enhanced Trust and Interpretability: Users and stakeholders, particularly in high-stakes domains such as journalism and policy-making, require a transparent measure of reliability. The Actuality score provides a systematic and interpretable way to assess the trustworthiness of AI-generated explanations. 3. Robust Generalization Across Domains: The reference-free nature of the metric allows it to generalize across diverse news topics, from politics and economics to science and health. This adaptability ensures that factual consistency is maintained regardless of domain-specific variations.

**Conclusion:** The Actuality score represents a critical advancement in factual evaluation for AI-generated news explanations. By addressing the limitations of reference-based metrics and enforcing stricter factual alignment, it provides a more reliable, scalable, and interpretable solution for ranking explanations. This metric strengthens the integrity of AI-driven summarization and reasoning systems, paving the way for more factually grounded and trustworthy AI applications in news generation and beyond.

## 4.2 Explaining *Finesse*

**Rationale:** In veracity prediction and explanation, model hallucination leads to inconsistent probability distributions for identical inputs, especially in ambiguous cases. To address this, we introduce the *Finesse* score, which quantifies variance in probability distributions, directly measuring model uncertainty. By integrating this parameter, we ensure uncertain cases are appropriately scaled, enhancing the model's reliability.

Additionally, journalistic writing naturally exhibits stylistic variance even within the same organization. The *Finesse* score captures this real-world diversity by incorporating varied temperature settings, ensuring explanations remain consistent while reflecting natural writing fluctuations. This balance improves both factual robustness and alignment in AI-generated explanations.

**Calculation:** The *Finesse* parameter was computed by generating five responses for each preferred explanation using a high-temperature value of 0.9, ensuring controlled diversity while maintaining coherence. Given these five output distributions, we calculated the variance for each token across them and then averaged these variances to derive the final *Finesse* score. This score quantifies the model's output uncertainty and was integrated into the DPO loss function to scale log probabilities, prioritizing explanations with higher consistency and lower uncertainty, thereby enhancing factual reliability. We show the computation of the *Finesse* term below.

The *Finesse* Score is computed by measuring the variance in model predictions across multiple decoding temperatures. Specifically, given a response distribution $P(y \mid x, T)$ at temperature $T$, we define the variance:

$$v = \frac{1}{n} \sum_{i=1}^{n} \left( P(y_i \mid x, T_i) - \bar{P}(y \mid x) \right)^2 \tag{1}$$

where $\bar{P}(y \mid x)$ represents the mean probability distribution across sampled temperatures. The parameter $\epsilon$ ensures numerical stability.

By integrating this score into the Hin-DPO formulation, we dynamically adjust preference-based ranking to account for inherent uncertainty in model predictions.

**Justification:** The choice of a high-temperature value each of the five times strikes a balance between introducing diversity and maintaining response quality, as verified through empirical tests. Additionally, the computational cost of generating five responses was manageable for our dataset size.

## 4.3 Modified DPO Loss Function

This section presents the modifications made to the DPO loss function by incorporating domain-specific parameters such as *Actuality* and *Finesse* scores. These additions aim to balance factual accuracy and uncertainty, ensuring that the model generates reliable and contextually relevant outputs aligned with human preferences.

$$L_{\text{Hin-DPO}}(\pi_\theta; \pi_{\text{ref}}) = -\mathbb{E}_{(x, y_w, y_l) \sim D} \left[ \log \sigma \left( \beta \cdot S(x, y_w, y_l) \right) \right] \tag{2}$$

$$S(x, y_w, y_l) = \frac{(1 + s_w) \log \left( \frac{\pi_\theta(y_w | x)}{\pi_{\text{ref}}(y_w | x)} \right) - \max(0.01, s_l) \log \left( \frac{\pi_\theta(y_l | x)}{\pi_{\text{ref}}(y_l | x)} \right)}{v + \epsilon} \tag{3}$$

Furthermore, we present below the algorithmic description of our Hin-DPO training

---

**Algorithm 2** Our Hin-DPO Training Algorithm

---

**Require:** Training dataset ad Dataloader $D$ with win and lose samples (paired or unpaired), initial model parameters $\theta_0$, reference model $\pi_{\text{ref}}$, number of iterations $T$, scaling factor $\beta$, temperature parameter $\tau$

1: Initialize model $\pi_\theta$ with parameters $\theta_0$
2: Set $\pi_\theta$ to training mode and $\pi_{\text{ref}}$ to evaluation mode
3: **for** iteration $= 1$ to $T$ **do**
4:   **for** each batch in $D$ **do**
5:     Initialize running mean $\mu \leftarrow 0$ and running variance $\sigma^2 \leftarrow 0$ {Running statistics for probability distribution}
6:     Set num_iter $= 5$ {Number of iterations for variance computation}
7:     **for** iter $= 1$ to num_iter **do**
8:       Compute logits for the preferred response:
9:       logits $\leftarrow \pi_\theta(\text{pref\_ids}, \text{pref\_mask}).\text{logits}$
10:       Compute probabilities:
11:       probs $\leftarrow \exp(\log\_\text{probs}(\text{logits, pref\_ids}))$
12:       Update Mean and Variance:
13:       $\mu \leftarrow \mu + \frac{\text{probs} - \mu}{\text{iter}}$
14:       $\sigma^2 \leftarrow \sigma^2 + (\text{probs} - \mu) \times (\text{probs} - \mu)$
15:     **end for**
16:     Compute final variance:
17:     $\sigma^2 \leftarrow \frac{\sigma^2}{\text{num\_iter} - 1}$
18:     Get log probabilities for preferred and dispreferred responses using $\pi_\theta$:
19:     model_pref_log $\leftarrow \log\_\text{prob}(\pi_\theta(\text{pref\_ids}, \text{pref\_mask}), \text{pref\_ids})$
20:     model_dispref_log $\leftarrow \log\_\text{prob}(\pi_\theta(\text{dispref\_ids}, \text{dispref\_mask}), \text{dispref\_ids})$
21:     Get log probabilities for preferred and dispreferred responses using reference model $\pi_{\text{ref}}$:
22:     ref_pref_log $\leftarrow \log\_\text{prob}(\pi_{\text{ref}}(\text{pref\_ids}, \text{pref\_mask}), \text{pref\_ids})$
23:     ref_dispref_log $\leftarrow \log\_\text{prob}(\pi_{\text{ref}}(\text{dispref\_ids}, \text{dispref\_mask}), \text{dispref\_ids})$
24:     Compute Hin-DPO loss:
25:     loss $\leftarrow$ Hin-DPO_loss(model_pref_log, model_dispref_log, ref_pref_log, ref_dispref_log, $\sigma^2, \beta$)
26:     Backpropagate loss:
27:     loss.backward()
28:     Update model parameters:
29:     $\theta \leftarrow$ optimizer.step()
30:   **end for**
31: **end for**

---

In Equation 2 & 3, $L_{\text{Hin-DPO}}$ represents the modified Direct Preference Optimization (DPO) loss function. The terms $\pi_\theta(y \mid x)$ and $\pi_{\text{ref}}(y \mid x)$ denote the probabilities assigned by the learned policy and the reference policy, respectively, for a response $y$ given input $x$. The dataset sample $(x, y_w, y_l) \sim D$ consists of an input $x$, a preferred response $y_w$, and a rejected response $y_l$. The *Actuality* scores $s_w$ and $s_l$ quantify the factual

accuracy of the preferred and rejected responses, respectively, ensuring factually grounded optimization. The hyperparameter $\beta$ controls the preference weighting, while $v$ represents the *Finesse* score, capturing prediction variance to regulate uncertainty. A small positive term $\epsilon$ is added to prevent division by zero, and $\sigma(\cdot)$ denotes the sigmoid function for smooth preference modelling. The gradient analysis of *Hin-DPO* is presented in Section A.8 of the appendix.

**Preserving Core DPO Training Process:** Despite the inclusion of these additional parameters, the core preference-based alignment in DPO remains intact. The *Actuality* and *Finesse* scores act as dynamic weighting mechanisms, adjusting rewards to improve factual consistency while mitigating uncertainty.

**Utilization of *Actuality*:** We leverage *Actuality* scores to modulate the weighting of preferred and rejected responses. Specifically, $(1 + s_w)$ amplifies the log probability of the preferred response, ensuring that factually accurate explanations are prioritized. Conversely, $\max(0.01, s_l)$ is applied to the rejected response, penalizing factually weak explanations without over-penalization.

**Impact of *Finesse*:** The *Finesse* score, computed via variance in predictions across different temperature settings, regulates the scaling factor in DPO loss. Low variance (indicating stable predictions) amplifies the reward, while high variance (close to 1) retains standard DPO behaviour. A small learnable parameter $\epsilon$ prevents instability.

# 5    Experimental Setup

**LLMs and PLMs Used:**  We fine-tuned five models, comprising three Large Language Models (LLMs)—Gemma-2-9B-It (Team, 2024), Llama-3.1-8B-Instruct (Dubey et al., 2024), and Mistral-7B-Instruct-v0.3 (Jiang et al., 2023) and two Pre-trained Language Models (PLMs): mBART-large-50 (Tang et al., 2020) and mT5-large (Xue et al., 2021). **Evaluation Metrics:** The quality of the generated explanations was assessed using three key metrics: BERTSCORE (Zhang et al., 2019), ROUGE-1,2, L score (Lin, 2004) and METEOR score (Banerjee & Lavie, 2005). Given the involvement of Hindi, we utilized the Polyglot tokenizer (Al-Rfou' et al., 2013) to evaluate ROUGE-1, 2, L, and METEOR scores. The hyperparameters are presented in the appendix (Section A.1).

Table 1: Performance comparison across models. **Abbreviations:** R-1: ROUGE-1, R-2: ROUGE-2, R-L: ROUGE-L, MT: METEOR, BS: BERTScore, Act: Actuality, Fin: Finesse. Bold values denote the best performance.

| Model → | mBART | | | | | mT5 | | | | | Gemma2-9B | | |
|---|---|---|---|---|---|---|---|---|---|---|---|---|---|
| Config↓ | R-1 | R-2 | R-L | MT | BS | R-1 | R-2 | R-L | MT | BS | R-1 | R-2 | R-L |
| Base | 13.03 | 6.12 | 9.72 | 18.23 | 69.94 | 15.13 | 6.84 | 9.93 | 19.16 | 70.14 | 28.17 | 17.36 | 21.86 |
| Base+SFT | 13.33 | 7.00 | 9.97 | 18.92 | 70.13 | 16.12 | 7.01 | 9.94 | 19.42 | 71.64 | 29.01 | 17.76 | 22.51 |
| DPO | 13.96 | 7.02 | 10.25 | 18.98 | 72.12 | 16.24 | 7.45 | 10.69 | 19.63 | 72.29 | 30.11 | 18.72 | 23.41 |
| DPO+Act | 14.10 | 6.92 | 10.45 | 19.25 | 73.07 | 16.31 | 7.42 | 10.81 | 19.95 | 73.06 | 30.83 | 19.01 | 24.15 |
| DPO+Fin | 13.71 | 6.15 | 10.29 | 19.09 | 72.50 | 16.11 | 7.31 | 10.74 | 19.66 | 72.30 | 30.25 | 18.87 | 23.58 |
| Hin-DPO | **14.66** | **7.12** | **10.77** | **19.73** | **74.01** | **16.40** | **7.58** | **11.12** | **20.03** | **74.19** | **31.12** | **19.78** | **24.68** |

| Model → | Mistral-7B | | | | | Llama3.1-8B | | | | | Gemma2-9B | |
|---|---|---|---|---|---|---|---|---|---|---|---|---|
| Config↓ | R-1 | R-2 | R-L | MT | BS | R-1 | R-2 | R-L | MT | BS | MT | BS |
| Base | 23.01 | 13.24 | 19.44 | 23.12 | 71.11 | 29.32 | 17.27 | 22.56 | 27.45 | 74.24 | 27.68 | 72.12 |
| Base+SFT | 24.51 | 13.78 | 20.21 | 23.99 | 72.77 | 31.01 | 17.87 | 23.16 | 27.79 | 74.99 | 28.19 | 74.19 |
| DPO | 24.76 | 14.02 | 20.78 | 25.04 | 73.14 | 31.15 | 18.68 | 24.17 | 28.71 | 75.23 | 29.17 | 76.11 |
| DPO+Act | 24.95 | 14.45 | 20.94 | 26.11 | 75.17 | 32.41 | 19.25 | 24.79 | 29.45 | 76.18 | 29.84 | 78.73 |
| DPO+Fin | 24.79 | 14.11 | 20.79 | 25.12 | 73.87 | 31.78 | 18.76 | 24.43 | 28.98 | 76.02 | 29.20 | 76.56 |
| Hin-DPO | **26.21** | **15.04** | **21.93** | **26.55** | **76.24** | **33.89** | **20.04** | **25.72** | **30.47** | **76.96** | **31.74** | **80.02** |

# 6 Results and Analysis

In this section, we present our findings and provide a comprehensive analysis of how our proposed framework, **DeFactoX**, compares against a range of baseline approaches. Table 1 illustrates the effectiveness of our method across multiple evaluation metrics, demonstrating its ability to generate coherent, contextually relevant, and factually accurate Hindi news explanations.

A key observation from the results is that incorporating DPO-based techniques (*DPO*, *DPO+Actuality*, and *DPO+Finesse*) consistently enhances model performance. This directly addresses **RQ1** by showcasing improvements in coherence, contextual relevance, and factual accuracy, all of which are critical for reliable news summarization. Among these approaches, *Hin-DPO* achieves the best overall performance, highlighting the benefits of language-specific fine-tuning. This finding underscores the necessity of tailored approaches for resource-constrained languages like Hindi, aligning with our investigation in **RQ2**.

Additionally, the consistent improvements across multiple evaluation metrics, including ROUGE, METEOR, and BERTSCORE, further emphasize that fine-tuning with DPO techniques enhances semantic alignment, fluency, and contextual accuracy. The particularly high BERTSCORE results achieved by *Hin-DPO* provide strong evidence of its ability to generate human-like explanations that are both contextually grounded and comprehensible, effectively addressing **RQ3**. This demonstrates that leveraging curriculum learning alongside DPO-based training can systematically refine a model's factual correctness over time, enabling a scalable strategy for misinformation mitigation.

**Real-Life Applicability:** The advancements achieved by *DeFactoX*, particularly with the *Hin-DPO* variant, have direct implications for real-world applications. Media houses can integrate our framework into their content verification pipelines to ensure that news articles remain factually accurate and contextually coherent before publication, while fact-checking organizations can leverage it to automate the detection of misleading narratives, enabling faster identification and correction of the misinformation. Additionally, government agencies and NGOs working on misinformation mitigation can deploy our approach to enhance digital literacy and promote reliable news dissemination, especially in Hindi-speaking regions. Beyond institutional use, *DeFactoX* benefits end-users by providing contextually accurate explanations, improving their ability to discern truth from misinformation. This is particularly crucial for rural and semi-urban populations where Hindi is the primary language of news consumption, and existing automated systems struggle with maintaining factual consistency.

# 7 Conclusion & Limitations

In conclusion, this work presents a novel framework, **DeFactoX**, that effectively addresses the challenges of misinformation detection and explanation generation in Hindi. By creating a synthetic Hindi preference dataset and leveraging advanced techniques like DPO, factscore integration, and curriculum learning, we have demonstrated a scalable method for generating coherent, contextually relevant, and factually accurate explanations. While our approach enhances automated misinformation detection for Hindi, certain challenges remain. Our framework relies on high-quality annotated data sourced from fact-checking websites for ground truth, which can be limited for low-resource languages. Additionally, the performance is influenced by the capabilities of current pre-trained LLMs, which may face difficulties with highly complex topics. Future work could explore multilingual transfer techniques to leverage resources from high-resource languages. Moreover, incorporating user feedback in a human-in-the-loop setting could further refine explanation quality and factual alignment.

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

# A Appendix

## A.1 Major Hyperparameters

This section outlines the resources and parameters used in our experiments. Dataset generation with Mistral was conducted on an NVIDIA RTX 3090 GPU (24 GB), while GPT and Gemini relied on smaller GPUs, as their API-based tasks required minimal computational resources. Fine-tuning of LLMs utilized an L40S GPU with 45 GB memory for efficient processing. The dataset was split into 75% training, 5% validation, and 20% testing. For data generation, GPT and Mistral used a temperature of 0.7, top_k of 50, and top_p of 0.95, while Gemini ran with default parameters. The Direct Preference Optimization (DPO) alignment involved 10 epochs, a learning rate of 1e-4, a batch size of 2, and a beta value of 0.6. Alignment took 48 hours, with model fine-tuning conducted in parallel.

## A.2 Prompt for Standardizing True News Explanations

To ensure uniformity in true news explanations, we employed **prompt engineering** (Sahoo et al., 2024) with the **GPT-4o-mini model** (Achiam et al., 2023). The goal was to make true news explanations explicitly state their veracity, aligning them with the structured reasoning found in fake news explanations.

Below is the prompt used for this task:

> **Prompt for Explicit True News Reasoning**
>
> *I will provide you with an article containing a verified news story along with related information. Your task is to rewrite the article as an explanation, explicitly emphasizing that the news is true and using only the content provided in the article to substantiate this claim. You must not remove or add any information beyond what is provided. Avoid using extra newline characters and adhere to the format of the input article. It is mandatory to add sentences that emphasize that the news is true, and include such sentences multiple times if possible. Restrict the output to Hindi language only.*

This prompt ensures:

1. **Consistency:** True news explanations explicitly affirm their veracity, aligning them with fake news explanations.
2. **Factual Integrity:** The process avoids introducing biases while preserving the original content.
3. **Linguistic Uniformity:** All explanations are generated in Hindi for consistency.

Examples of standardized explanations are provided below.

## A.3 Example of Standardizing True News Explanations

To demonstrate the effectiveness of our standardized prompting approach, we provide a concrete example shown in Figure 3 that shows both the input (original verified news article) and the output (standardized explanation) generated using our carefully crafted prompt with GPT-4o-mini.

**Input (Original Article):** The input consists of a verified news story about Prime Minister Narendra Modi's speech during the 2024 Lok Sabha elections campaign. The article highlights his address in Andhra Pradesh, where he shared the stage with key regional leaders and outlined BJP's plans, emphasizing NDA's growing strength and future aspirations. The content contains factual elements such as:

**Event Context:** Election Commission announcing the 2024 Lok Sabha election schedule.

**Location and Participants:** PM Modi's rally in Andhra Pradesh with TDP Chief Chandrababu Naidu and Jana Sena Chief Pawan Kalyan.

**Key Messages:** PM Modi's speech highlighting BJP-NDA's progress. "400+ seats" slogan. Positive outlook on NDA's influence and regional unity.

**Specific Quotes:** Emphasis on progress, veracity, and development themes mentioned in the speech. The input, as shown in Figure 3 (left side), is presented in Hindi and remains free of any explicit reasoning of its veracity.

**Output (Standardized Explanation):** The output, shown in Figure 3 (right side), demonstrates how the original article is rewritten using the standardized prompt described in Section 4.1. The following refinements are evident in the output:

**Explicit Affirmation of Truthfulness:** The output explicitly emphasizes multiple times that the news is true. For instance:

This news is completely true.

There is no false or misleading information in this report.

Such affirmations ensure the reader is repeatedly reassured about the authenticity of the content. **Retention of Content Integrity:** All factual details, including event context, quotes, and location, are retained without omission or addition. The content remains faithful to the input, preserving its original integrity.

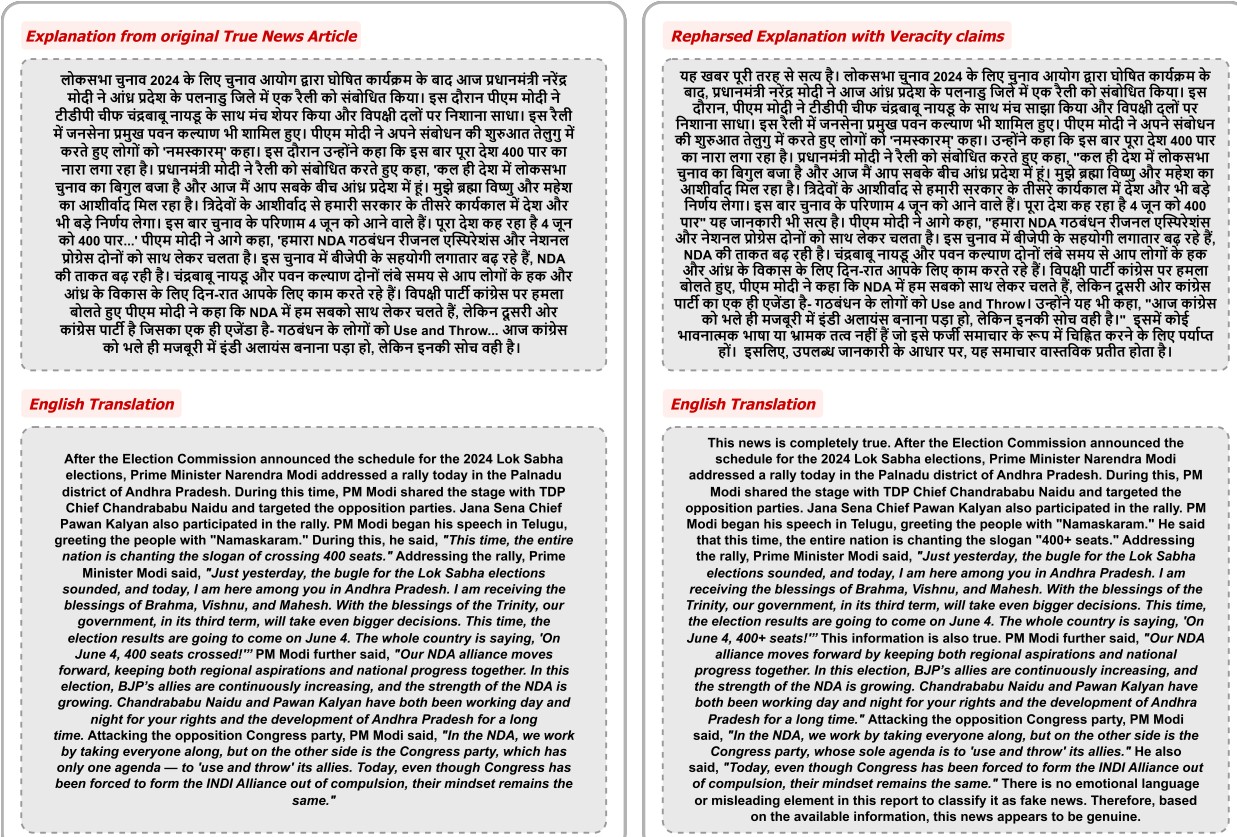

Figure 3: Snippet of True news transformation.

**Coherent Restructuring:** While no new information is introduced, the content is made clearer and more coherent through slight restructuring. For example, the verification statements are seamlessly interwoven with the original facts to provide a more structured explanation.

**Language Uniformity:** The output adheres strictly to the Hindi language, as per the prompt's requirements, ensuring consistency across all generated explanations.

**Repetition for Reinforcement:** Sentences affirming the news' truth are repeated strategically to reinforce its veracity. For instance: This news is true and based on factual information.

## A.4 Prompt for Generating Rejected Responses

To generate rejected responses, we prompted the LLMs to classify a given news article as either fake or real and provide a justification in a structured paragraph. The prompt was carefully designed to assess the models' reasoning capabilities while ensuring the responses remained comparable to real-world explanations. Below is the full prompt used in our experiments:

---

**Prompt for News Explanation**

*Task: I will provide you with a news article. Your task is to do the following:*
*1. Predict whether the news is fake or real.*
*2. Provide a detailed explanation for your prediction.*
*Ensure your response is written as a flowing paragraph, avoiding bullet points, numbering, or any other structured format. The explanation should naturally justify your prediction, without adding any extraneous information.*
*Here is the news article:* `article`
*Answer in the form of a detailed paragraph.*

---

This prompt ensured that the LLMs generated structured responses while remaining consistent in their reasoning. However, these responses were treated as the negative class since they lacked explicit grounding in verified fact-checking methods.

## A.5 Theoretical Foundation of DPO

Given a text sequence (commonly referred to as a prompt) $x$, a sequence $y = [y_1, y_2, \ldots, y_N]$ is generated as a response to the prompt $x$. An autoregressive language model $\pi$, when provided with the prompt $x$, can generate the response sequence $y$ following the probability decomposition:

$$\pi(y|x) = \prod_{t=1}^{N} \pi(y_t|x, y_{<t}),$$

where $y_{<t}$ denotes the preceding tokens in the response sequence. Given a preference dataset $D = \{(x, y_w, y_l)\}$, where $y_w$ and $y_l$ are the winning (preferred) response and losing (less preferred) response, respectively, while $x$ is the given prompt.

A well-known preference alignment technique is Reinforcement Learning from Human Feedback (RLHF). It can be divided into two parts. A reward model $r_\phi$ is first trained using the Bradley-Terry model (Bradley & Terry, 1952):

$$P(y_w \succ y_l|x) = \frac{\exp(r_\phi(x, y_w))}{\exp(r_\phi(x, y_w)) + \exp(r_\phi(x, y_l))}$$

After obtaining the reward model $r_\phi$, the second step is to use Proximal Policy Optimization (PPO) (Schulman et al., 2017) to optimize the language model $\pi_\theta$, so that the model's output has a higher reward score, as shown in the following training objective:

$$\max_{\pi_\theta} \mathbb{E}_{x \sim D, y \sim \pi_\theta(\cdot|x)} \left[ r_\phi(x, y) - \beta D_{KL}(\pi_\theta \| \pi_{ref}) \right],$$

where $D_{KL}$ measures the divergence between $\pi_\theta$ and $\pi_{ref}$ (initial model). Rafailov et al. (2024b) mathematically derived the optimal policy $\pi_\theta^*$ from the reward model $r(x, y)$ as follows:

$$\pi^*(y|x) = \frac{1}{Z(x)}\pi_{ref}(y|x)e^{r_\phi(x,y)},$$

where $Z(x)$ is the partition function. We could easily get $r_\phi(x, y) = \beta \log \frac{\pi^*(y|x)}{\pi_{ref}(y|x)} - Z(x)$ from Eq. 3. Substituting into the Bradley-Terry model yields the DPO objective:

$$L_{\text{DPO}}(\pi_\theta; \pi_{\text{ref}}) = -\mathbb{E}_{(x,y_w,y_l)\sim D}\left[\log \sigma\left(\Delta\right)\right],$$

where

$$\Delta = \beta \cdot \log\left(\frac{\pi_\theta(y_w \mid x)}{\pi_{\text{ref}}(y_w \mid x)}\right) - \beta \cdot \log\left(\frac{\pi_\theta(y_l \mid x)}{\pi_{\text{ref}}(y_l \mid x)}\right).$$

## A.6   Theoretical Foundation of Curry-DPO

Curry-DPO builds upon the Direct Preference Optimization (DPO) framework by incorporating curriculum learning and multiple preference pairs. The key theoretical aspects are:

### A.6.1   Multiple Preference Pairs

Given a prompt $p$, Curry-DPO utilizes multiple responses $R_1, R_2, R_3, R_4$ with varying quality ratings, where $R_1 > R_2 > R_3 > R_4$ in terms of preference. This allows for the creation of multiple preference pairs $(R_i, R_j)$ where $i < j$, expanding the training data beyond the single pair used in standard DPO.

### A.6.2   Curriculum Learning

Curry-DPO orders these preference pairs from "easy" to "hard" based on the difference in quality ratings. The curriculum $C$ can be defined as:

$$\begin{aligned} C = \{&(R_1, R_4), (R_1, R_3), (R_1, R_2), \\ &(R_2, R_4), (R_2, R_3), (R_3, R_4)\} \end{aligned} \tag{4}$$

where pairs with larger rating differences are considered easier and presented earlier in training.

### A.6.3   Iterative Training

The training process involves multiple iterations, where for iteration $t$:

$$\theta_t = \text{DPO}(\theta_{t-1}, (Y_w, Y_l)_t) \tag{5}$$

Here, $\theta_t$ represents the model parameters at iteration $t$, and $(Y_w, Y_l)_t$ is the preference pair from curriculum $C$ used in that iteration.

### A.6.4 Reference Model Updates

Curry-DPO updates the reference model after each iteration, unlike standard DPO:

$$\theta_{\mathrm{ref},t} = \theta_{t-1} \tag{6}$$

This allows the model to gradually adapt to more challenging preference pairs.

### A.6.5 Optimization Objective

The core DPO objective is maintained, minimizing the following loss:

$$\mathcal{L}_{\mathrm{DPO}}(\theta) = -\mathbb{E}_{(x,y_w,y_l)\sim\mathcal{D}}\left[\log\sigma(r_\theta(x,y_w) - r_\theta(x,y_l))\right] \tag{7}$$

where $r_\theta(x,y)$ is the reward assigned by the model to response $y$ for prompt $x$.

This theoretical framework allows Curry-DPO to leverage multiple preference pairs and curriculum learning, leading to improved alignment and performance on various benchmarks compared to standard DPO.

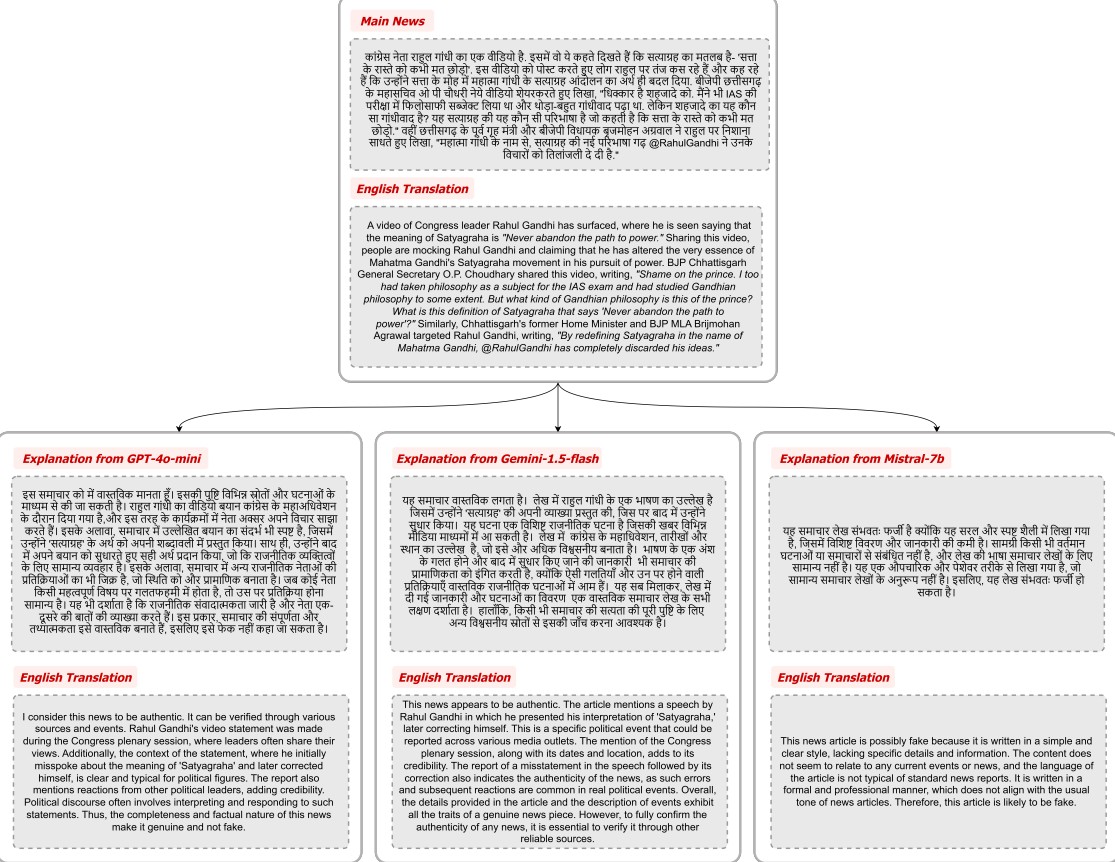

Figure 4: Snippet of Non-Preferred Response Generation.

## A.7    Example of Non-Preferred Outputs

To illustrate the composition of the synthetic preference dataset, we provide an example via Figure 4 showcasing:

**Input:** An unbiased news article about Congress leader Rahul Gandhi's speech. The news discusses a viral video featuring Congress leader Rahul Gandhi, where he is seen saying, "Satyagraha means – never leave the path of power."

This statement sparked controversy, and critics, particularly from the BJP, have mocked Rahul Gandhi, accusing him of distorting the meaning of Mahatma Gandhi's Satyagraha movement.

Key reactions include: 1. O.P. Chaudhary, BJP Chhattisgarh General Secretary, shared the video and criticized Rahul Gandhi, saying: - "Shame on him. I studied philosophy for the IAS exam, and I know Gandhian principles. But what kind of Gandhian philosophy is this? What kind of Satyagraha definition says never leave the path of power?"

2. Brijmohan Agrawal, former Chhattisgarh Home Minister and BJP MLA, also criticized Rahul Gandhi, stating: - "Rahul Gandhi has created a new definition of Satyagraha in Mahatma Gandhi's name, abandoning his ideals."

The controversy centres around the alleged misrepresentation of Mahatma Gandhi's philosophy, with political opponents using the incident to attack Rahul Gandhi and question his understanding of Gandhian values.

**Non-Preferred Outputs (Machine-Generated Explanations)** The non-preferred outputs, shown in the bottom section of Figure 4, are generated by feeding the first paragraph of the original article into the prompt described in Section A.4. These outputs were produced using three large language models (GPT-4o-mini, Mistral-7B-v0.1, and Gemini-1.5-Flash), and their key deficiencies are summarized below:

**Model 1: GPT-4o-mini**

The explanation provided is a non-preferred response due to its lack of precision and depth. While it asserts that the news is authentic, it fails to directly address the core issue of misinformation—the selective editing of Rahul Gandhi's speech. A preferred explanation would highlight that the viral video was intentionally cropped to exclude his immediate correction, which distorted the meaning of his statement. Additionally, the response does not cite credible fact-checking sources or provide verifiable evidence, relying instead on vague generalizations like "typical for political figures."

Furthermore, the explanation lacks a critical analysis of how the manipulated video gained traction through political reactions. It mentions reactions from leaders but does not contextualize their role in amplifying the misinformation. A stronger explanation would focus on identifying the manipulation, referencing fact-checking timelines, and clarifying the corrected statement, ensuring a clear, fact-based debunking of the viral claim. This clarity and specificity are essential for effective misinformation detection.

**Model 2: Mistral-7B-v0.1**

The explanation provided lacks relevance and critical engagement with the content of the news article. It categorizes the article as "fake" based purely on the writing style, stating that its "simple and clear style" and "formal and professional" tone are indicators of inauthenticity. However, these are not valid criteria for determining the veracity of a news article. News articles can be written in various styles, and a formal tone is common in credible journalism.

The response also fails to address the core issue: the misleading nature of the viral video and the misinterpretation of Rahul Gandhi's statement. A more robust explanation would directly focus on the distortion of Gandhi's words and the intentional cropping of his speech. Furthermore, the explanation does not engage with any available fact-checking resources or offer a logical argument for why the content is fake, instead relying on subjective and unsupported claims about the article's tone and style. This lack of critical reasoning makes the explanation inadequate for identifying and debunking misinformation thus falling into non-preferred output.

**Model 3: Gemini-1.5-Flash**

The model provides an explanation that is generally more balanced, but still contains several issues in the reasoning process (thus, a non-preferred response). While the response acknowledges the political event and Rahul Gandhi's misstatement, it fails to critically evaluate the core of the misinformation—specifically, how the video was edited to distort Gandhi's words. The response states that the misstatement followed by a correction indicates authenticity, but this overlooks the key issue that the video in question was edited in a way that intentionally misrepresents the original message, a crucial detail for identifying misinformation.

Additionally, the model's mention of "verifying through other reliable sources" is a positive note, but the explanation doesn't elaborate on how this verification process should be conducted or mention specific fact-checking sources or methods. A more accurate response would have focused on the nature of the edited video, the political motivations behind the distortion, and the importance of consulting fact-checking websites to confirm the claims made. This would have provided a more nuanced approach to distinguishing between authentic and misleading news.

## A.8 Gradient Analysis of Hin-DPO

### A.8.1 Objective Function

The Hin-DPO objective is given by:

$$L_{\text{Hin-DPO}}(\pi_\theta; \pi_{\text{ref}}) = -\mathbb{E}_{(x,y_w,y_l)\sim\mathcal{D}} \left[ \log \sigma \left( \frac{\beta\left((1+s_w)r_w - \max(0.01, s_l)r_l\right)}{v + \epsilon} \right) \right], \tag{8}$$

where

$$r_w = \log \frac{\pi_\theta(y_w|x)}{\pi_{\text{ref}}(y_w|x)}, \quad r_l = \log \frac{\pi_\theta(y_l|x)}{\pi_{\text{ref}}(y_l|x)}. \tag{9}$$

### A.8.2 Loss function formulation

$$r(x, y) = \beta \log \frac{\pi_\theta(y|x)}{\pi_{\text{ref}}(y|x)} + \beta \log Z(x)$$

In our method we use two different type reward modelling for Preferred and rejected response.

For Preferred response

$$r_w(x, y) = \beta(1 + s_w) \log \frac{\pi_\theta(y|x)}{\pi_{\text{ref}}(y|x)} + \beta \log Z(x)$$

For Rejected response

$$r_l(x, y) = \beta \max(0.01, s_l) \log \frac{\pi_\theta(y|x)}{\pi_{\text{ref}}(y|x)} + \beta \log Z(x)$$

$$P(y_w \succ y_l \mid x) = \frac{\exp(r_w(x, y_w))}{\exp(r_w(x, y_w)) + \exp(r_l(x, y_l))} = \frac{1}{1 + \exp(r_l(x, y_l) - r_w(x, y_w))}$$

$$P(y_w \succ y_l \mid x) = \frac{1}{1 + \exp\left(\beta \max(0.01, s_l) \log \frac{\pi_\theta(y_l|x)}{\pi_{\text{ref}}(y_l|x)} - \beta(1 + s_w) \log \frac{\pi_\theta(y_w|x)}{\pi_{\text{ref}}(y_w|x)}\right)}$$

$$P(y_w \succ y_l \mid x) = \sigma\left(\beta(1 + s_w) \log \frac{\pi_\theta(y_w \mid x)}{\pi_{\text{ref}}(y_w \mid x)} - \beta \max(0.01, s_l) \log \frac{\pi_\theta(y_l \mid x)}{\pi_{\text{ref}}(y_l \mid x)}\right)$$

### A.8.3 Gradient Derivation

Let $u$ denote the argument of the sigmoid function:

$$u = \frac{\beta\left((1 + s_w)r_w - \max(0.01, s_l)r_l\right)}{v + \epsilon}. \tag{10}$$

The gradient of $L_{\text{Hin-DPO}}$ is:

$$\nabla_\theta L_{\text{Hin-DPO}} = -\mathbb{E}_{(x,y_w,y_l)\sim\mathcal{D}}\left[(1 - \sigma(u))\nabla_\theta u\right]. \tag{11}$$

Compute $\nabla_\theta u$ as:

$$\nabla_\theta u = \frac{\beta}{v + \epsilon}\left[(1 + s_w)\nabla_\theta r_w - \max(0.01, s_l)\nabla_\theta r_l\right]. \tag{12}$$

Substituting $\nabla_\theta u$ back, the gradient becomes:

$$\nabla_\theta L_{\text{Hin-DPO}} = -\mathbb{E}_{(x,y_w,y_l)\sim\mathcal{D}}\left[\frac{\beta^2\sigma(u)}{v + \epsilon}\left[(1 + s_w)\nabla_\theta r_w - \max(0.01, s_l)\nabla_\theta r_l\right]\right]. \tag{13}$$

The gradient formulation for Hin-DPO is a natural extension of the standard DPO framework, tailored for preference learning in domains requiring factual grounding and uncertainty control. By incorporating the *Actuality* scores $s_w$ and $s_l$, the gradient aligns with domain-specific factual quality, amplifying the influence of more factually accurate preferred responses via $(1 + s_w)$ and softening the impact of rejected responses through $\max(0.01, s_l)$. This asymmetric treatment ensures that the learning process emphasizes factual correctness while avoiding over-penalization of partially informative generations. Additionally, scaling by the *Finesse* term $v + \epsilon$ allows the model to adjust updates based on prediction variance, ensuring smoother convergence across varying confidence levels. Overall, the derived gradient preserves the probabilistic preference modelling of DPO while injecting interpretable and learnable domain-specific signals, resulting in a more robust and fact-aware policy optimization process.

