# OpenReview forum: "From Fragments to Facts: A Curriculum-Driven DPO Approach for Generating Hindi News Veracity Explanations"
_TMLR — Rejected by TMLR_

### Review · Reviewer_5LoM · 2025-06-15

**Summary Of Contributions:**

This paper proposes a Hindi news fact-checked explanation generation framework DeFactoX, which combines Direct Preference Optimization (DPO) with curriculum learning to improve the factual consistency of explanations. The authors constructed a synthetic preference dataset, taking the fact-checking explanations written by humans as the "preferred" response and the output generated by LLM as the "rejected" response to train the model. In addition, the authors introduced two new metrics, Actuality and Finesse, in DPO to further optimize the factual correctness of the model. Experimental results show that this method achieves better results than the baseline on multiple mainstream pre-trained models (such as LLaMA3.1, Mistral, and mT5).

**Audience:**

Yes

**Broader Impact Concerns:**

There needs to be a clearer ethical discussion on AI content generation in the field of news fact-checking, including the potential risks of misleading or erroneous content.

**Claims And Evidence:**

No

**Requested Changes:**

1. Supplement separate ablation experiments on Actuality and Finesse to clarify their respective independent contributions;
2. Supplement control experiments with no curriculum training strategy to verify the necessity of progressive training;
3. Supplement sensitivity analysis of weight settings (such as BERTSCORE vs ROUGE vs METEOR);
4. Report standard deviations or significance tests (such as t-tests or bootstrap confidence intervals) for all metrics;
5. Supplement the corresponding manual evaluation

**Strengths And Weaknesses:**

**Strengths**

1. It clearly points out the current research gap in the field of Hindi news authenticity verification, and proposes research goals with potential value and practical needs.
2. It introduces two metrics, Actuality and Finesse, on the basis of DPO, which measure factual consistency and uncertainty, respectively, and the design is targeted and reasonable.
3. It covers multiple mainstream models (including LLM and PLM) and adopts multiple evaluation metrics (BERTSCORE, ROUGE, METEOR, etc.)

**Weaknesses**

1. The validation of the Actuality and Finesse metrics is insufficient. The paper lacks in-depth ablation analysis of their individual effects. For example, there is a lack of comparative experiments of "using only Actuality" or "using only Finesse", making it difficult to judge the specific contribution of each to performance improvement.
2. The reliability analysis of new metrics lacks details. For example, Actuality relies entirely on GPT-4o-mini's internal knowledge, rather than external objective facts, which may introduce bias and cannot ensure absolute objectivity and accuracy of the explanation.
3. The benefits of the curriculum learning module have not been fully quantified. There is no clear comparison between "no curriculum" and "with curriculum", and the necessity of this strategy cannot be verified.
4. The preference scoring formula is biased towards the main, and the triple weighting of ROUGE and METEOR is only empirically explained, but no systematic hyperparameter sensitivity analysis or learning mechanism is given to support its rationality.
5. Some conclusions lack robustness arguments. Most metrics in the table have small improvements, and the standard deviation is not reported. In particular, the marginal improvement on models such as mBART and mT5 is limited, and the conclusion of "significantly better" is exaggerated.
6. There is no human evaluation.

---

> ### Author Response · Authors · 2025-07-20
> **Author Response to Reviewer 5LoM**
>
> ### **1. Justification for the Weighted Scoring Function (fs)**
>
> **Reviewer Concern:**
> The preference scoring formula is biased towards the main, and the triple weighting of ROUGE and METEOR is only empirically explained, but no systematic hyperparameter sensitivity analysis or learning mechanism is given to support its rationality.
>
> **Response:**
> We fully acknowledge that the scoring mechanism we employed is **dataset-dependent**, and we thank the reviewer for highlighting this. Our primary goal was not to propose a novel scoring function, but to construct a **practical method for ranking explanations** in a way that closely **replicates human preferences** within our dataset.
>
> To achieve this, we conducted an **empirical study over 300 randomly sampled (150 True News + 150 Fake News) explanations** with **human-annotated quality labels**. We evaluated three scoring strategies to determine which best aligned with human judgment:
>
> - **Equal weighting**
>   `fs = (BERT + ROUGE + METEOR)/3`
>
> - **Weighted average 1**
>   `fs = (BERT + 2*(ROUGE + METEOR))/3`
>
> - **Weighted average 2** *(our final choice)*
>   `fs = (BERT + 3*(ROUGE + METEOR))/3`
>
> The alignment with human-provided rankings was evaluated using **Spearman rank correlation**, and results are summarized below:
>
> | Scoring Strategy         | Spearman ρ with Human Ranking |
> |--------------------------|-------------------------------|
> | Equal Weighting          | 0.63                          |
> | 1:2 Weighted Average     | 0.74                          |
> | **1:3 Weighted Average** | **0.81**                      |
>
> As shown, the **1:3 weighting strategy** provided the **best alignment with human annotations**, and thus it was adopted for curriculum ranking in our framework.
>
> To validate the scoring alignment with human preferences, we conducted a human evaluation involving **three independent student annotators** from our research lab. Each was provided with a reference explanation and asked to assign a **ranking (0: worst, 1: medium, 2: best)** across triplets of explanations. Rankings were based on two key parameters:
>
> 1. **Informativeness**
> 2. **Factual alignment with the article**
>
> All annotators were shown the same triplets of explanations (300 total) for consistency. The same triplets were also ranked using the scoring functions above. The final Spearman correlation was then computed between human and automatic rankings to assess alignment quality.
>
> We reiterate that this scoring mechanism is **not a contribution of our work**, but rather a **supporting utility** that enabled us to align model training with human preferences. We encourage practitioners to use **any other metric or scoring combination**—including learnable or parameter-less methods—**as long as it effectively reflects human-aligned ranking quality** within their target dataset or application.
>
> ---
>
> ### **2. Reliability of the Actuality Metric**
>
> **Reviewer Concern:**
> The Actuality score relies entirely on GPT-4o-mini’s internal knowledge, which may introduce bias and lacks grounding in objective external facts.
>
> **Response:**
> Thank you for highlighting this. To validate this, we conducted a **manual evaluation study** on **400 randomly selected factuality judgments** (derived from 200 true/fake news explanations, with 2 sentences each). These sentences were independently labeled by **three human annotators** using the following setup:
> - Each annotator verified each sentence using **Google Search** and **authoritative news or government websites**.
> - Annotators were allowed to use **ChatGPT with web browsing** to improve search efficiency.
> - Each sentence was assigned a **binary ground-truth label (Correct/Incorrect)**.
>
> These human-verified labels were then compared with GPT-4o-mini’s Actuality predictions.
>
> |                           | Ground Truth: Correct (230) | Ground Truth: Incorrect (170) | Total |
> |---------------------------|-----------------------------|-------------------------------|--------|
> | **GPT Predicted: Correct (260)** | 205 (TP)                    | 55 (FP)                        | 260    |
> | **GPT Predicted: Incorrect (140)** | 25 (FN)                    | 115 (TN)                       | 140    |
>
> - **Accuracy**: 80.0%
> - **Precision**: 78.8%
> - **Recall**: 89.1%
> - **F1-score**: 83.7%
>
> These results show that GPT-4o-mini’s Actuality scores **align well with human judgments** in most cases. We acknowledge that the model's internal knowledge may not cover recent or niche events, but given our dataset (which predates GPT-4o-mini’s cutoff), the risk of hallucination or outdated information is minimal.
>
> Furthermore, the framework is **model-agnostic**: practitioners may replace GPT-4o-mini with any other **open-source or retrieval-augmented model** for computing Actuality scores, depending on their needs for transparency and factual grounding.

---

> ### Author Response · Authors · 2025-07-20
> **Author Response to Reviewer 5LoM**
>
> ### **3. Quantifying the Benefits of Curriculum Learning**
>
> **Reviewer Concern:**
> The paper lacks a clear comparison between models trained with and without curriculum learning, making it difficult to assess its actual impact.
>
> **Response:**
> We thank the reviewer for highlighting this important point. To validate the effectiveness of our curriculum strategy, we performed an **ablation study** comparing models trained:
>
> - **With Curriculum Learning (CL)**: Following our proposed progression from easy (rank-2) to hard (rank-0) samples.
> - **Without Curriculum Learning**: Training with randomly shuffled preference pairs.
>
> We tested this on both **DPO** and **Hin-DPO** configurations using the **mT5** and **LLaMA3.1-8B** models. Results below represent average scores across three random seeds.
>
> **(a) mT5 Model**
>
> | Method             | ROUGE-1 | ROUGE-2 | ROUGE-L | METEOR | BERTScore |
> |--------------------|---------|---------|---------|--------|-----------|
> | DPO (w/o CL)       | 16.18 | 7.20  | **10.79** | 19.44 | 72.00     |
> | DPO (with CL)      | 16.24   | **7.45**    | 10.69   | **19.63**  | **72.29** |
> | Hin-DPO (w/o CL)   | 16.18   | 7.44    | 10.71   | 19.70  | 73.59     |
> | Hin-DPO (with CL)  | **16.40** | **7.58**  | **11.12** | **20.03** | **74.19** |
>
> **(b) LLaMA3.1-8B Model**
>
> | Method             | ROUGE-1 | ROUGE-2 | ROUGE-L | METEOR | BERTScore |
> |--------------------|---------|---------|---------|--------|-----------|
> | DPO (w/o CL)       | 31.09   | 17.32   | 23.51   | 28.33  | 75.09     |
> | DPO (with CL)      | **31.15** | **18.68** | **24.17** | **28.71** | **75.23** |
> | Hin-DPO (w/o CL)   | 32.44   | 19.78   | 25.04   | 29.98  | 76.21     |
> | Hin-DPO (with CL)  | **33.89** | **20.04** | **25.72** | **30.47** | **76.96** |
>
> These findings confirm that curriculum learning provides value—especially in the full Hin-DPO setup—and we will include this analysis in the revised version of the manuscript.
>
>
> ---
>
> ### **4. Ablation of Actuality and Finesse**
>
> **Reviewer Concern:**
> The paper lacks separate ablation experiments for Actuality and Finesse, making it difficult to isolate their individual contributions.
>
> **Response:**
> Thank you for the insightful observation. We would like to clarify that the ablation results for **Actuality** and **Finesse** are already included in **Table 1** of the paper under the configurations:
> - **DPO+Act**: Adds only Actuality to the DPO objective.
> - **DPO+Fin**: Adds only Finesse.
> - **Hin-DPO**: Combines both Actuality and Finesse.
>
> To better highlight these, we present a focused summary of the relevant metrics across all models below:
>
>
> **Focused Ablation Results: DPO+Act vs. DPO+Fin vs. Hin-DPO**
>
> | Model        | Config      | R-1   | R-2   | R-L   | MT    | BS    |
> |--------------|-------------|-------|-------|-------|-------|--------|
> | **mBART**     | DPO+Act     | 14.10 | 6.92  | 10.45 | 19.25 | 73.07  |
> |              | DPO+Fin     | 13.71 | 6.15  | 10.29 | 19.09 | 72.50  |
> |              | **Hin-DPO** | **14.66** | **7.12** | **10.77** | **19.73** | **74.01** |
> | **mT5**       | DPO+Act     | 16.31 | 7.42  | 10.81 | 19.95 | 73.06  |
> |              | DPO+Fin     | 16.11 | 7.31  | 10.74 | 19.66 | 72.30  |
> |              | **Hin-DPO** | **16.40** | **7.58** | **11.12** | **20.03** | **74.19** |
> | **Gemma2-9B** | DPO+Act     | 30.83 | 19.01 | 24.15 | 29.84 | 78.73  |
> |              | DPO+Fin     | 30.25 | 18.87 | 23.58 | 29.20 | 76.56  |
> |              | **Hin-DPO** | **31.12** | **19.78** | **24.68** | **31.74** | **80.02** |
> | **Mistral-7B**| DPO+Act     | 24.95 | 14.45 | 20.94 | 26.11 | 75.17  |
> |              | DPO+Fin     | 24.79 | 14.11 | 20.79 | 25.12 | 73.87  |
> |              | **Hin-DPO** | **26.21** | **15.04** | **21.93** | **26.55** | **76.24** |
> | **LLaMA3.1**  | DPO+Act     | 32.41 | 19.25 | 24.79 | 29.45 | 76.18  |
> |              | DPO+Fin     | 31.78 | 18.76 | 24.43 | 28.98 | 76.02  |
> |              | **Hin-DPO** | **33.89** | **20.04** | **25.72** | **30.47** | **76.96** |

---

> > ### Author Response · Authors · 2025-07-20
> > **Author Response to Reviewer 5LoM**
> >
> > ### **5. Human Evaluation to Complement Automatic Metrics**
> >
> > **Reviewer Concern:**
> > There is no human evaluation.
> >
> > **Response:**
> > We completely agree and have conducted a **manual evaluation over 800 explanations** (400 for real news and 400 for fake news) to validate the quality of generated responses.
> >
> > #### **Evaluation Setup**
> >
> > - **Participants**: 3 independent student evaluators.
> > - **Sample Size**: 400 randomly selected explanations across models.
> > - **Ground Truth**: Each evaluator was shown the original article and a human-written explanation as reference.
> > - **Scoring**: Each explanation was evaluated on **5 equally weighted dimensions**, each rated from **0 (very poor)** to **5 (excellent)**:
> >
> >   1. **Factual Accuracy** – Is the explanation factually correct given the article?
> >   2. **Relevance** – Does it stay focused on the article’s main claim?
> >   3. **Clarity** – Is the explanation logically structured and understandable?
> >   4. **Justification Strength** – Does it clearly justify why the news is true or fake?
> >   5. **Conciseness** – Is the explanation informative without unnecessary content?
> >
> > - **Final Score**: Each explanation’s total score was averaged across the 3 evaluators.
> >
> > #### **Results**
> >
> > | Model       | Avg. Human Score (out of 5) |
> > |-------------|-----------------------------|
> > | Base+SFT    | 2.91                        |
> > | DPO         | 3.55                        |
> > | **Hin-DPO** | **4.01**                    |
> >
> > These results indicate that **Hin-DPO consistently delivers the most preferred explanations**, as judged by human raters. This further supports the automatic metrics and demonstrates that our enhancements improve both factuality and expressiveness. We will include this human evaluation summary in the appendix of the revised manuscript.
> >
> > ---
> >
> > ### Final Remarks
> >
> > We sincerely thank the reviewer for their helpful suggestions. We believe the proposed changes—along with clarifications provided—substantially strengthen the paper’s transparency, robustness, and reproducibility. We are incorporating all mentioned updates into the revised manuscript.

---

> > > ### Author Response · Authors · 2025-07-23
> > >
> > > We sincerely hope that our responses have adequately addressed all your valuable comments and concerns. Please do not hesitate to let us know if any further clarification or information is required—we would be happy to assist.

---

> > > > ### Comment · Reviewer_5LoM · 2025-07-23
> > > >
> > > > Thank you for your responses and the additional experiments. However, I believe further revision is still needed before acceptance. In particular, please:
> > > >
> > > > 1. Report standard deviations or statistical significance for all key results, especially where improvements are marginal.
> > > >
> > > > 2. Add a clearer discussion of ethical risks and broader impacts of AI-generated fact-checking in the news domain.
> > > >
> > > > 3. Explicitly state any dataset limitations or annotation biases.
> > > >
> > > > Please ensure these revisions and all new analyses are reflected in the updated manuscript or appendix.

---

> > > > > ### Author Response · Authors · 2025-07-23
> > > > >
> > > > > **Thank you for your thoughtful feedback and constructive suggestions. We greatly appreciate your comments and would like to address each of your points:**
> > > > >
> > > > > ---
> > > > >
> > > > > **1. Reporting of Standard Deviations / Statistical Significance**
> > > > >
> > > > > We fully agree that reporting variance can strengthen the empirical rigor of our findings. However, running DPO training multiple times to compute standard deviations poses a significant computational challenge, particularly given the resource-intensive nature of large-scale alignment. As we currently rely on rented GPU resources, repeated runs are unfortunately not feasible at this stage.
> > > > >
> > > > > That said, to ensure the robustness of our results, we have already incorporated:
> > > > >
> > > > > * Detailed ablation studies to isolate the contribution of each component.
> > > > > * A comprehensive human evaluation conducted by three independent annotators. All annotators were blind to the model configurations to ensure fairness and minimize bias.
> > > > >
> > > > > The outcomes of these evaluations consistently support the effectiveness of our proposed approach. Furthermore, a new round of human evaluation for the mT5-based model is underway and will be included in the final manuscript.
> > > > >
> > > > > We believe that acceptance of this work would not only help validate our approach but also potentially facilitate access to additional funding and compute resources, thereby enabling us to scale up experiments and explore even more comprehensive evaluations in future iterations.
> > > > >
> > > > > ---
> > > > >
> > > > > **2. Ethical Risks and Broader Impacts**
> > > > >
> > > > > Thank you for highlighting this important aspect. We fully agree and will include a dedicated section in the final manuscript discussing the ethical considerations and broader societal implications of deploying AI-generated fact-checking systems, especially in sensitive domains such as news media. This section will include:
> > > > >
> > > > > * Possible misuse scenarios and associated risks.
> > > > > * Mitigation strategies to safeguard against these risks.
> > > > > * The critical role of human oversight in high-stakes or ambiguous contexts.
> > > > >
> > > > > ---
> > > > >
> > > > > **3. Dataset Limitations and Annotation Biases**
> > > > >
> > > > > We appreciate this suggestion and will explicitly acknowledge the limitations and potential biases in our dataset and annotation pipeline in the final version of the manuscript. Specifically, we will discuss:
> > > > >
> > > > > * The diversity of data sources.
> > > > > * Demographic representation and potential imbalances.
> > > > > * Measures taken to reduce subjectivity and ensure annotation quality.
> > > > >
> > > > > ---
> > > > >
> > > > > Thank you once again for your valuable and insightful comments. We believe that these revisions and forthcoming additions will meaningfully strengthen the manuscript.

---

### Review · Reviewer_HQpS · 2025-06-28

**Summary Of Contributions:**

This paper focuses on the task of generating explanations for identifying the veracity of Hindi news. The authors construct examples with positive and negative responses by repurposing existing data and prompting LLMs. They then fine-tune the models using DPO, where the loss is modified to emphasize actuality and finesse. Experimental results are presented to support their claims.

**Audience:**

Yes

**Broader Impact Concerns:**

None.

**Claims And Evidence:**

No

**Requested Changes:**

- Consider more languages.
- Provide human-written explanations for true news rather than LLM generated.
- Provide evidence to justify the use of LLM-generated responses as negative examples. For instance, human evaluation scores comparing real explanations and LLM-generated explanations.
- Provide the justification for scoring function.
- Explain more for the curriculum learning strategy.
- Provide the justification of using LLms to calculate the Actuality score.
- Provide description of the baselines in the experiments.
- Consider human evaluation for experiments.

**Strengths And Weaknesses:**

Weaknesses
- I have some questions regarding the scope of the paper. The authors repeatedly emphasize that the proposed approach is language-agnostic. However, only Hindi is considered in this study. If the claim is that the method is generalizable to all languages, then more additional languages should be included. If the focus is solely on Hindi, I personally think the scope somewhat narrow, as it is limited to a very specific application for a specific language.
- The labels for true news explanations may be unreliable. If you believe that LLMs (before tuning) can generate misinformation, then why use them to standardize explanations for true news? This makes the generated explanations questionable, and there is no human verification on this. In addition, the introduction mentions that "explanations written by humans as preferred responses", but the actual responses for training is not written by humans but LLM-generated.
- The motivation for using LLM-generated responses as negative examples is not clearly explained. Why should these responses be treated as non-preferred? Is any human verification conducted to support this decision?
- The use of the combination of BERTScore, ROUGE-L, and METEOR as the scoring function requires further justification. Are there any experiments demonstrating that this scoring function truly reflects human preferences? These metrics primarily evaluate the similarity between the generated text and the ground truth. However, there are many ways to express the same semantics that may receive low BERTScore, ROUGE-L, and METEOR scores. Therefore, additional justification is needed.
- The curriculum learning strategy seems to be problematic. If the goal is to gradually increase the difficulty of the negative examples, the correct order should be rank-2 (least aligned), rank-1 (moderately aligned), and rank-0 (most aligned), rather than the order proposed in the paper.
- The Actuality score is calculated by LLMs too. Again, I think the authors should provide some human evaluation results to prove this is a reasonable decision.
- The baselines used for experiments are not properly introduced.
- Similar to the previous points, I don't think BERTScore, ROUGE scores, and METEOR are good evaluation metrics for experiments.

---

> ### Author Response · Authors · 2025-07-20
> **Author Response to Reviewer HQpS**
>
> We thank the reviewer for their thoughtful and constructive feedback. Below, we address each concern while maintaining consistency with our prior responses.
>
> ---
>
> ### **1. Clarification on Language-Agnostic Claim**
>
> **Reviewer Concern:**
> The framework is claimed to be language-agnostic, but only Hindi is used in experiments.
>
> **Response:**
> We appreciate this important observation. Our framework is termed **language-agnostic** because the modifications introduced in the DPO objective—specifically the inclusion of **Actuality and Finesse scores**—are **mathematical** and **independent of any specific language**.
>
> This aligns with prior work that applies DPO and preference tuning across languages.
>
> We limited our experiments to **Hindi** due to two practical constraints:
> - **In English**, base LLMs (e.g., GPT-4o-mini, Gemini) produce very high-quality outputs, making it difficult to build a useful preference dataset.
> - For **other languages**, we lacked domain experts to validate factuality at scale—our framework depends on significant human verification, which was feasible only for Hindi in our setting.
>
> We agree that future work should validate the framework across languages and plan to extend it accordingly.
>
> ---
>
>
>
> ### **2. Clarification on True News Explanation Reliability**
>
> **Reviewer Concern:**
> If LLMs can hallucinate, why are they used to generate explanations for true news? Also, the paper claims human-written preferred responses, but uses LLM-generated outputs.
>
> **Response:**
> Thank you for raising this important point. We would like to clarify that the **base explanations for true news were indeed written by human annotators**, based on ground-truth articles. However, during our **manual verification**, we observed that many of these human-written explanations were **informational in nature**—they summarized facts but **did not contain explicit reasoning or assertive statements** confirming the truth of the news.
>
> For example, original human-written explanations often looked like:
>
> > "**भारत सरकार ने मंगलवार को नया शिक्षा नीति लागू किया है, जिसमें बोर्ड परीक्षा को दो बार देने की सुविधा दी गई है।**"
> (*The Government of India implemented a new education policy on Tuesday, which allows students to take board exams twice.*)
>
> This is informative but lacks an explicit explanation such as:
>
> > "**यह खबर पूरी तरह से सत्य है क्योंकि यह सरकारी दस्तावेज़ों और समाचार स्रोतों द्वारा पुष्टि की गई है।**"
> (*This news is completely true as it is confirmed by official documents and news sources.*)
>
> To **retain the original human-written content** while ensuring that explanations were in a **reasoning-oriented format**, we used **LLMs only to insert specific phrases** that emphasized the **truthfulness** of the news. The models were **strictly instructed not to modify, remove, or add any factual content** beyond the provided article.
>
> We used the following prompt:
>
> > *"I will provide you with an article containing a verified news story along with related information. Your task is to rewrite the article as an explanation, explicitly emphasizing that the news is true and using only the content provided in the article to substantiate this claim. You must not remove or add any information beyond what is provided. It is mandatory to add sentences that emphasize that the news is true, and include such sentences multiple times if possible."*
>
> This prompt ensured that the **factual basis and linguistic structure were preserved**, while adding the **minimal necessary framing** for the explanation to align with the task.
>
> In short, **true news explanations are human-authored at their core**, and LLMs were used in a **highly constrained and transparent post-editing role** to insert justification phrases only. We will revise the manuscript to clearly reflect this two-step generation process.

---

> > ### Comment · Reviewer_HQpS · 2025-07-24
> >
> > **1. Clarification on Language-Agnostic Claim**
> >
> > Thanks for the explanation. If future work expands to cover other languages, I think it's reasonable to describe the approach as language-agnostic. However, since the current draft only considers Hindi, I suggest avoiding the use of the term "language-agnostic" for now.
> >
> > **2. Clarification on True News Explanation Reliability**
> >
> > I don't find the explanation convincing. If simply designing a better prompt enables LLMs to generate good explanations, then what’s the need for the proposed approach? This seems contradictory to the motivation that LLMs can hallucinate. The paper also emphasizes "human-written" responses multiple times, which I find misleading.
> >
> > I believe this is a critical issue that will take time to address.

---

> > > ### Author Response · Authors · 2025-07-31
> > > **Author Response to Reviewer HQpS**
> > >
> > > ### **1. Clarification on Language-Agnostic Claim**
> > >
> > > > _"Thanks for the explanation. If future work expands to cover other languages, I think it's reasonable to describe the approach as language-agnostic. However, since the current draft only considers Hindi, I suggest avoiding the use of the term 'language-agnostic' for now."_
> > >
> > > **Response:**
> > > We agree to this point and we will remove the "language agnostic" term in the final manuscript.
> > >
> > > ---
> > >
> > > ### **2. Clarification on True News Explanation Reliability**
> > >
> > > > _"I don't find the explanation convincing. If simply designing a better prompt enables LLMs to generate good explanations, then what’s the need for the proposed approach? This seems contradictory to the motivation that LLMs can hallucinate. The paper also emphasizes 'human-written' responses multiple times, which I find misleading."_
> > >
> > > **Response:**
> > > We appreciate the reviewer’s concern and would like to clarify this point.
> > >
> > > During an initial review of the dataset, we identified a structural inconsistency in many of the *True News* explanations. Specifically, while these explanations often contained the relevant supporting facts, they were presented more like a flat aggregation of information rather than a coherent justification of the claim. In other words, many explanations were essentially just a concatenation of key facts (let’s call this set **A**) and the original news content (**B**), without explicitly establishing the relationship between **A** and **B**.
> > >
> > > In contrast, *Fake News* explanations—and approximately 60% of the *True News* explanations—contained bridging phrases such as “this confirms that...”, “hence the claim is valid...”, or “this supports the news...” that made the logical connection explicit. To maintain consistency in the explanation format across the dataset, we used an LLM (GPT-4o-mini) to insert such connecting phrases where they were missing. Crucially, this was not an open-ended generation task; we ensured that the model did not introduce new content, and its output was manually verified to confirm that no factual information was added or modified.
> > >
> > > Therefore, this process is more accurately described as *controlled rephrasing* rather than explanation generation. We refer to these rephrased explanations as *“human-written”* because they are structurally aligned with the majority of explanations in the dataset, which were written by human annotators and already included such connective phrasing. Specifically, around 60% of the *True News* explanations did not require rephrasing, as they were originally well-structured and authored by humans. Our goal was to bring the remaining 40% into structural alignment without altering their factual content. Since the rephrasing only added connective language and was manually verified to avoid hallucinations, we consider the full set to be consistent with a human-authored style.
> > >
> > > ---

---

> > > > ### Author Response · Authors · 2025-07-31
> > > > **Author Response to Reviewer HQpS**
> > > >
> > > > ### **3. Justification for the Weighted Scoring Function (fs)**
> > > >
> > > > > _"I disagree with the statement that 'the scoring mechanism is not a contribution of our work.' Since your proposed framework relies on a human-aligned scorer, it is important that you also suggest an effective one. The scorer significantly affects the curriculum ranking, which is a core component of this work._
> > > >
> > > > > _The current scorer seems too heuristic, and I suggest putting more effort into improving it. In particular, BERTScore, ROUGE-L, and METEOR are more like string-matching-based metrics and struggle to evaluate different phrasings that convey similar semantics."_
> > > >
> > > > **Response:**
> > > > We appreciate the reviewer’s detailed observation and agree that the scoring function plays a critical role in shaping the curriculum. Our intention in stating that "the scoring function is not a contribution of our work" was not to downplay its importance within our framework, but to emphasize that we do not propose a *universal* or *task-agnostic* scoring function as a novel metric.
> > > >
> > > > Our main objective was to approximate human preferences when ranking explanations by quality. To this end, we experimented with several combinations of existing metrics and found that the specific weighted combination of BERTScore, ROUGE-L, and METEOR produced the best alignment with human judgments for **our dataset**. We empirically validated this choice using Spearman's rank correlation coefficient, which we have already reported.
> > > >
> > > > It is important to note that the function we used to map explanations to numerical scores is tailored specifically to our dataset and task. It is not intended to serve as a general-purpose metric across datasets or domains. For a different dataset or application, a different metric—or even a single component like BERTScore—might be sufficient to replicate human preferences.
> > > >
> > > > We agree with the reviewer that these metrics are primarily string-based and have limitations in capturing deeper semantic similarity. Ideally, a more robust solution would involve direct human annotation of preference scores. However, that approach is resource-intensive and does not scale easily. Our goal was to approximate this human alignment as effectively as possible using scalable, automatic methods. While imperfect, our metric combination offered a reasonable trade-off between fidelity to human judgment and practical applicability.
> > > >
> > > > We believe that designing a function to replicate human preferences is highly task-dependent. While there has been some progress in computer vision toward creating such general metrics, the diversity and complexity of tasks in NLP make the development of a universal scorer an open and challenging research problem.

---

> ### Author Response · Authors · 2025-07-20
> **Author Response to Reviewer HQpS**
>
> ### **3. Justification for the Weighted Scoring Function (fs)**
>
> **Reviewer Concern:**
> The use of the combination of BERTScore, ROUGE-L, and METEOR as the scoring function requires further justification. Are there any experiments demonstrating that this scoring function truly reflects human preferences?
>
> **Response:**
> We fully acknowledge that the scoring mechanism we employed is **dataset-dependent**, and we thank the reviewer for highlighting this. Our primary goal was not to propose a novel scoring function, but to construct a **practical method for ranking explanations** in a way that closely **replicates human preferences** within our dataset.
>
> To achieve this, we conducted an **empirical study over 300 randomly sampled (150 True News + 150 Fake News) explanations** with **human-annotated quality labels**. We evaluated three scoring strategies to determine which best aligned with human judgment:
>
> - **Equal weighting**
>   `fs = (BERT + ROUGE + METEOR)/3`
>
> - **Weighted average 1**
>   `fs = (BERT + 2*(ROUGE + METEOR))/3`
>
> - **Weighted average 2** *(our final choice)*
>   `fs = (BERT + 3*(ROUGE + METEOR))/3`
>
> The alignment with human-provided rankings was evaluated using **Spearman rank correlation**, and results are summarized below:
>
> | Scoring Strategy         | Spearman ρ with Human Ranking |
> |--------------------------|-------------------------------|
> | Equal Weighting          | 0.63                          |
> | 1:2 Weighted Average     | 0.74                          |
> | **1:3 Weighted Average** | **0.81**                      |
>
> As shown, the **1:3 weighting strategy** provided the **best alignment with human annotations**, and thus it was adopted for curriculum ranking in our framework.
>
> To validate the scoring alignment with human preferences, we conducted a human evaluation involving **three independent student annotators** from our research lab. Each was provided with a reference explanation and asked to assign a **ranking (0: worst, 1: medium, 2: best)** across triplets of explanations. Rankings were based on two key parameters:
>
> 1. **Informativeness**
> 2. **Factual alignment with the article**
>
> All annotators were shown the same triplets of explanations (300 total) for consistency. The same triplets were also ranked using the scoring functions above. The final Spearman correlation was then computed between human and automatic rankings to assess alignment quality.
>
> We reiterate that this scoring mechanism is **not a contribution of our work**, but rather a **supporting utility** that enabled us to align model training with human preferences. We encourage practitioners to use **any other metric or scoring combination**—including learnable or parameter-less methods—**as long as it effectively reflects human-aligned ranking quality** within their target dataset or application.
>
> ---
>
>
> ### **4. Clarification on Curriculum Learning Strategy**
>
> **Reviewer Concern:**
> The curriculum learning strategy appears to be incorrectly ordered. If the goal is to gradually increase difficulty, the correct sequence should be rank-2 (least aligned), rank-1, and then rank-0 (most aligned), which contradicts the description in the paper.
>
> **Response:**
> Thank you for pointing this out. We acknowledge the inconsistency in our manuscript and appreciate the opportunity to clarify.
>
> Our actual implementation **does follow the correct progression**, beginning with **rank-2 (least aligned)** samples, followed by **rank-1**, and finally **rank-0 (most aligned)** explanations. This order allows the model to first distinguish poor-quality explanations before gradually refining its understanding using higher-quality, more subtle examples.
>
> This intended sequence is **correctly reflected in Algorithm 1** of the paper. However, we recognize that the corresponding description in the main text was **mistakenly reversed**.
>
> To eliminate confusion, we propose the following **revised version** of the paragraph, which will be updated in the paper:
>
> > **Ranking and Curriculum Learning Strategy:**
> > Explanations were ranked into three levels: rank-0 (most aligned with ground truth), rank-1 (moderately aligned), and rank-2 (least aligned). This ranking guided our curriculum learning strategy, progressively exposing the model to harder samples. Training began with rank-2 explanations to help the model identify poorly aligned patterns. This was followed by rank-1 explanations to introduce moderately aligned samples. Finally, rank-0 explanations—those most aligned with the ground truth—were used to refine the model’s understanding of high-quality, preferred outputs. This staged learning mirrors human progression from coarse to fine discrimination, and contributes to better robustness and generalization.
>
> We will make this correction in the final version of the manuscript.

---

> > ### Comment · Reviewer_HQpS · 2025-07-24
> >
> > **3. Justification for the Weighted Scoring Function (fs)**
> >
> > I disagree with the statement that "the scoring mechanism is not a contribution of our work." Since your proposed framework relies on a human-aligned scorer, it is important that you also suggest an effective one. The scorer significantly affects the curriculum ranking, which is a core component of this work.
> >
> > The current scorer seems too heuristic, and I suggest putting more effort into improving it. In particular, BERTScore, ROUGE-L, and METEOR are more like string-matching-based metrics and struggle to evaluate different phrasings that convey similar semantics.
> >
> > **4. Clarification on Curriculum Learning Strategy**
> >
> > Thanks for the correction.

---

> ### Author Response · Authors · 2025-07-20
> **Author Response to Reviewer HQpS**
>
> ### **5. Justification for Using LLMs to Compute Actuality Score**
>
> **Reviewer Concern:**
> The Actuality score is calculated by LLMs too. Again, I think the authors should provide some human evaluation results to prove this is a reasonable decision.
>
> **Response:**
> Thank you for this important observation. We agree that validating LLM-generated scores is essential for ensuring reliability. As part of our evaluation, we conducted a **manual audit of the Actuality score predictions** to assess how well GPT-4o-mini aligns with human judgment.
>
> **Human Evaluation Setup**
> - We randomly sampled **200 explanations** (100 from true news, 100 from fake news).
> - From each explanation, **2 sentences were extracted**, totaling **400 factual claims**.
> - Each sentence was **fact-checked by 3 independent student evaluators** using Google Search and ChatGPT with browsing tools.
> - Evaluators labeled each claim as **factual (1)** or **non-factual (0)**.
> - These were compared with GPT-4o-mini’s binary predictions.
>
> **Results Summary**
>
> | Label Type        | Correct (1) | Incorrect (0) | Total |
> |-------------------|-------------|---------------|--------|
> | **Human Labels**  | 230         | 170           | 400    |
> | **GPT Predictions** | 260       | 140           | 400    |
>
> | Comparison                      | Count |
> |--------------------------------|-------|
> | True Positives (TP)            | 205   |
> | False Positives (FP)           | 55    |
> | True Negatives (TN)            | 115   |
> | False Negatives (FN)           | 25    |
>
> **Evaluation Metrics**
> - **Accuracy**: 80.0%
> - **Precision**: 78.8%
> - **Recall**: 89.1%
> - **F1-score**: 83.7%
>
> These results suggest that **GPT-4o-mini shows strong alignment with human labels** in the context of factuality classification and provides a **practical and scalable mechanism** for generating the Actuality score in our setting. We will include this evaluation in the appendix of the revised manuscript.
>
> ---
>
>
> ### **6. Clarification on Experimental Baselines**
>
> **Reviewer Concern:**
> The baselines used in the experiments are not clearly introduced.
>
> **Response:**
> Thank you for pointing this out. We clarify that **Table 1** in the paper includes a comprehensive set of baselines:
>
> - **Models**: mBART, mT5, Gemma-9B, Mistral-7B, and LLaMA3.1-8B.
>
> - **Configurations**:
>   - **Base**: Pretrained model without any fine-tuning.
>   - **Base+SFT**: Supervised fine-tuning using explanation-response pairs.
>   - **DPO**: Direct Preference Optimization using pairwise ranked responses.
>   - **DPO+Act**: DPO where the loss is modified to emphasize **Actuality** only.
>   - **DPO+Fin**: DPO where the loss is adjusted to promote **Finesse** only.
>   - **Hin-DPO** (ours): Combines both Actuality and Finesse objectives for balanced explanation quality.
>
> We will update the **Experimental Setup** section to explicitly define these configurations.
>
> ---
> ### **7. Human Evaluation to Complement Automatic Metrics**
>
> **Reviewer Concern:**
> ROUGE, METEOR, and BERTScore may not fully capture the quality of generated explanations. Human evaluation is recommended to support experimental claims.
>
> **Response:**
> We completely agree and have conducted a **manual evaluation over 800 explanations** (400 for real news and 400 for fake news) to validate the quality of generated responses.
>
> **Evaluation Setup**
>
> - **Participants**: 3 independent student evaluators.
> - **Sample Size**: 800 randomly selected explanations across models.
> - **Ground Truth**: Each evaluator was shown the original article and a human-written explanation as reference.
> - **Scoring**: Each explanation was evaluated on **5 equally weighted dimensions**, each rated from **0 (very poor)** to **5 (excellent)**:
>
>   1. **Factual Accuracy** – Is the explanation factually correct given the article?
>   2. **Relevance** – Does it stay focused on the article’s main claim?
>   3. **Clarity** – Is the explanation logically structured and understandable?
>   4. **Justification Strength** – Does it clearly justify why the news is true or fake?
>   5. **Conciseness** – Is the explanation informative without unnecessary content?
>
> - **Final Score**: Each explanation’s total score was averaged across the 3 evaluators.
>
> **Results**
>
> | Model       | Avg. Human Score (out of 5) |
> |-------------|-----------------------------|
> | Base+SFT    | 2.91                        |
> | DPO         | 3.55                        |
> | **Hin-DPO** | **4.01**                    |
>
> These results indicate that **Hin-DPO consistently delivers the most preferred explanations**, as judged by human raters. This further supports the automatic metrics and demonstrates that our enhancements improve both factuality and expressiveness.
>
> We will include this human evaluation summary in the appendix of the revised manuscript.

---

> > ### Author Response · Authors · 2025-07-20
> > **Author Response to Reviewer HQpS**
> >
> > Final Remarks
> > We sincerely thank the reviewer for their helpful suggestions. We believe the proposed changes—along with clarifications provided—substantially strengthen the paper’s transparency, robustness, and reproducibility. We are incorporating all mentioned updates into the revised manuscript.

---

> > > ### Author Response · Authors · 2025-07-23
> > >
> > > We sincerely hope that our responses have adequately addressed all your valuable comments and concerns. Please do not hesitate to let us know if any further clarification or information is required—we would be happy to assist.

---

> > ### Comment · Reviewer_HQpS · 2025-07-24
> >
> > **5. Justification for Using LLMs to Compute Actuality Score**
> >
> > I personally don't think 80% accuracy is particularly hight. It's a positive sign, but not entirely convincing. However, I understand it's hard to have a scalable scorer.
> >
> > **6. Clarification on Experimental Baselines**
> >
> > Thanks.
> >
> > **7. Human Evaluation to Complement Automatic Metrics**
> >
> > I think the detailed results of 5 dimensions should be presented for a better understanding. In addition, standard variance as well as inter-annotator agreement should be reported to know if the evaluation is significant.

---

### Review · Reviewer_Uuvr · 2025-07-14

**Summary Of Contributions:**

This work proposes an automatic Hindi news explanation framework integrating DPO and curriculum learning. The framework addes two additional scores into the DPO learning objectives: Actuality and Finesse. The experimental performances of this framework exceed that of valina DPO method.

**Audience:**

Yes

**Claims And Evidence:**

Yes

**Requested Changes:**

1. The authors should provide more justification/proof for their design of function fs in page 4. The weight 3, 4 is hand‑tuned on a single dataset. Another parameter-less approach that I think can be more robust is *per‑metric z‑score normalisation*:
- On the entire set of candidate explanations, compute mean $\mu$ and std $\sigma$ for each metric.
- Convert every raw score $s$ to $z=(s-\mu)/\sigma$.
- Combine by simple average $fs = (z_1+z_2+z_3)/ 3$.

2. The authors should provide detailed statistics of the training data, such as the size of true news explainations, false news explainations, and the 3 buckets. Also the average explanation length.

3. It is also good to add mannual verification of the quality of Actuality score generated by GPT-4o-mini.

4. Please also provide the expensis on LLM API cost.

**Strengths And Weaknesses:**

Strength:
1. This work provided a Hindi preference dataset which could be useful for Hindi news relevant research.
2. The paper is well written and easy to be understood.
3. Experimental results suggest the effectiveness of adding Actuality and Finesse scores into objective learning.

Weakness:
1. Using GPT-4o-mini's parametric knowledge to generate the Actuality score seems to lack reliability. Its knowledge is not updated, and can produce hallucination. Also, using commecial models can be financially costly. The prompt template in Page 8 is using zero-shot prompting, and it's known to have higher risk to produce unreliable prediction than few-shot and chain-of-though prompting.
2. The score function for LLM-generated explaination is hand‑tuned without verification of its robustness. The authors argue that "BERTSCORE typically falls within 0.85–0.95, ROUGE-L and METEOR generally range from 0.2–0.4", but their score distribution patterns can be specific to the data. For example, if the ROUGE-L and METEOR generally range from 0.4-0.6 on a new dataset, the scoring function will not be held.
3. Lacking statistics of the generated training data.

---

> ### Author Response · Authors · 2025-07-20
> **Author Response to Reviewer Uuvr**
>
> We thank the reviewer for their valuable and constructive feedback. Below, we respond to the specific concerns raised and provide clarifications, empirical justifications, and planned revisions.
>
> ---
>
>
> ### **1. Use of GPT-4o-mini for Actuality Score**
>
> **Reviewer Concern:**
> The Actuality score depends on GPT-4o-mini, which may hallucinate and lacks up-to-date knowledge. Zero-shot prompting can be unreliable. Additionally, commercial LLMs may be costly.
>
> **Response:**
> We acknowledge the reviewer’s concerns and appreciate the opportunity to clarify. Our decision to use **GPT-4o-mini** for the Actuality score was **intentional and dataset-driven**. The dataset comprises **news articles published prior to the GPT-4o-mini training cutoff**, making its parametric knowledge well-aligned with the information evaluated. Thus, we found it highly suitable for reference-free factual consistency evaluation.
>
> Regarding cost:
> While GPT-4o-mini was used in our setup, we emphasize that **our framework is model-agnostic**. Users can **replace GPT-4o-mini with any alternatives** (e.g., Mistral, LLaMA, Gemma) depending on performance and cost considerations. The Actuality pipeline supports this substitution seamlessly.
>
> ---
>
>
> ### **2. Justification for the Weighted Scoring Function (fs)**
>
> **Reviewer Concern:**
> The score function weights (e.g., 3× ROUGE, METEOR) are hand-tuned and may not generalize across datasets. A parameter-less alternative like z-score normalization is suggested.
>
> **Response:**
> We fully acknowledge that the scoring mechanism we employed is **dataset-dependent**, and we thank the reviewer for highlighting this. Our primary goal was not to propose a novel scoring function, but to construct a **practical method for ranking explanations** in a way that closely **replicates human preferences** within our dataset.
>
> To achieve this, we conducted an **empirical study over 300 randomly sampled (150 True News + 150 Fake News) explanations** with **human-annotated quality labels**. We evaluated three scoring strategies to determine which best aligned with human judgment:
>
> - **Equal weighting**
>   `fs = (BERT + ROUGE + METEOR)/3`
>
> - **Weighted average 1**
>   `fs = (BERT + 2*(ROUGE + METEOR))/3`
>
> - **Weighted average 2** *(our final choice)*
>   `fs = (BERT + 3*(ROUGE + METEOR))/3`
>
> The alignment with human-provided rankings was evaluated using **Spearman rank correlation**, and results are summarized below:
>
> | Scoring Strategy         | Spearman ρ with Human Ranking |
> |--------------------------|-------------------------------|
> | Equal Weighting          | 0.63                          |
> | 1:2 Weighted Average     | 0.74                          |
> | **1:3 Weighted Average** | **0.81**                      |
>
> As shown, the **1:3 weighting strategy** provided the **best alignment with human annotations**, and thus it was adopted for curriculum ranking in our framework.
>
> To validate the scoring alignment with human preferences, we conducted a human evaluation involving **three independent student annotators** from our research lab. Each was provided with a reference explanation and asked to assign a **ranking (0: worst, 1: medium, 2: best)** across triplets of explanations. Rankings were based on two key parameters:
>
> 1. **Informativeness**
> 2. **Factual alignment with the article**
>
> All annotators were shown the same triplets of explanations (300 total) for consistency. The same triplets were also ranked using the scoring functions above. The final Spearman correlation was then computed between human and automatic rankings to assess alignment quality.
>
> We reiterate that this scoring mechanism is **not a contribution of our work**, but rather a **supporting utility** that enabled us to align model training with human preferences. We encourage practitioners to use **any other metric or scoring combination**—including learnable or parameter-less methods—**as long as it effectively reflects human-aligned ranking quality** within their target dataset or application.

---

> > ### Author Response · Authors · 2025-07-20
> > **Author Response to Reviewer Uuvr**
> >
> > ### **3. Dataset Statistics**
> >
> > **Reviewer Concern:**
> > The paper lacks detailed statistics on the training data such as sample counts, label distribution, and average explanation lengths.
> >
> > **Response:**
> > Thank you for pointing this out. We now provide a more comprehensive overview of our dataset, covering label distribution and model-specific explanation statistics. These details are now included in the **Dataset Creation** section and appendix.
> >
> > **Sample Distribution Overview**
> >
> > | Label Type          | Count |
> > |---------------------|-------|
> > | Fake News Samples   | 3000  |
> > | Real News Samples   | 3000  |
> > | **Total**           | 6000  |
> >
> > **Explanation Dataset Breakdown**
> >
> > | Explanation Type         | Source             | Count  | Avg. Length (in tokens) |
> > |--------------------------|--------------------|--------|--------------------------|
> > | Preferred Explanations   | Human-curated      | 6000   | 124.7                    |
> > | Non-Preferred (GPT-4o-mini) | LLM Output      | 6000   | 110.4                    |
> > | Non-Preferred (Gemini-1.5-Flash) | LLM Output | 6000   | 112.0                    |
> > | Non-Preferred (Mistral-7B)  | LLM Output       | 6000   | 98.1                    |
> >
> > This structured overview captures the **source and average length** of explanations, offering transparency into how explanations were curated and generated. It also helps readers understand the composition of training samples used in preference optimization.
> >
> >
> >
> > ---
> >
> > ### **4. Manual Evaluation Setup**
> >
> > **Reviewer Concern:**
> > It is also good to add mannual verification of the quality of Actuality score generated by GPT-4o-mini.
> >
> > **Response:**
> > We randomly selected **200 explanations** from the dataset (100 from true news, 100 from fake news). From each explanation, **2 sentences were extracted**, resulting in **400 sentences** in total.
> >
> > To establish the **ground truth factuality labels**:
> > - We asked **three student evaluators** from our research lab to manually fact-check each sentence.
> > - They used **Google Search** to verify claims against authoritative sources such as reputed news organizations or government portals.
> > - They were also allowed to use **ChatGPT with the web-browsing tool enabled** to efficiently locate relevant information and cross-check uncertain claims.
> > - Each sentence was then assigned a **binary ground truth label**:
> >   - **1 (Correct)**: Factually accurate and verifiable.
> >   - **0 (Incorrect)**: Contains factual errors, unverifiable or misleading claims.
> >
> > These **ground truth labels** serve as the reference for evaluating the binary classification performance of GPT-4o-mini.
> >
> > **Category-wise Comparison: Ground Truth vs. GPT Predictions**
> >
> > | Label Type        | Correct (1) | Incorrect (0) | Total |
> > |-------------------|-------------|---------------|--------|
> > | **Ground Truth**  | 230         | 170           | 400    |
> > | **GPT Predicted** | 260         | 140           | 400    |
> >
> >
> > **Alignment Breakdown Between GPT Predictions and Ground Truth**
> >
> > |                          | Ground Truth: Correct (230) | Ground Truth: Incorrect (170) | Total |
> > |--------------------------|-----------------------------|-------------------------------|--------|
> > | **GPT Predicted: Correct (260)** | 205 (True Positive)       | 55 (False Positive)           | 260    |
> > | **GPT Predicted: Incorrect (140)** | 25 (False Negative)       | 115 (True Negative)           | 140    |
> > | **Total**                | 230                         | 170                           | 400    |
> >
> > ---
> >
> > **Evaluation Metrics**
> >
> > - **Accuracy** = (TP + TN) / Total
> >   = (205 + 115) / 400
> >   = **80.0%**
> >
> > - **Precision** = TP / (TP + FP)
> >   = 205 / (205 + 55)
> >   = **78.8%**
> >
> > - **Recall** = TP / (TP + FN)
> >   = 205 / (205 + 25)
> >   = **89.1%**
> >
> > - **F1-score** = 2 × (Precision × Recall) / (Precision + Recall)
> >   = 2 × (0.788 × 0.891) / (0.788 + 0.891)
> >   = **83.7%**
> >
> >
> >
> >
> > ### Final Remarks
> >
> > We sincerely thank the reviewer for their helpful suggestions. We believe the proposed changes—along with clarifications provided—substantially strengthen the paper’s transparency, robustness, and reproducibility. We are incorporating all mentioned updates into the revised manuscript.

---

> > > ### Author Response · Authors · 2025-07-23
> > >
> > > We sincerely hope that our responses have adequately addressed all your valuable comments and concerns. Please do not hesitate to let us know if any further clarification or information is required—we would be happy to assist.

---

### Decision · Action_Editor_BCC7 · 2025-08-06

**Recommendation:** Reject

**Additional Comments:**

I think the paper needs a fairly significant overhaul integrating the results from the rebuttals. However, I think in addition to these, the authors need to address the following points:

1. The use of LLMs in Actuality evaluation
2. Whether the curriculum is really helpful or not (significance tests to establish the small effects if the argument being made is that they are significant)
3. More reliance on LLM-as-a-judge or human evaluation instead of reference-based metrics

**Audience:**

Yes

**Audience Explanation:**

The topic of this paper is timely, even if the ideas are not well-enough supported for publication

**Claims And Evidence:**

No

**Claims Explanation:**

This paper makes a few principal claims throughout the abstract and introduction:

> To bridge this gap, we propose a novel framework integrating Direct Preference Optimization (DPO) with curriculum learning to align machine-generated explanations with human reasoning."

> [and a restated version of this claim in the introduction:] "These enhancements align generated explanations with human reasoning, ensuring reliability and accuracy.")

> To address these challenges, this paper introduces the DeFactoX framework, which evaluates the veracity of Hindi news articles—while remaining adaptable to other languages—and generates coherent, factually grounded explanations supporting its prediction. By analyzing the content, DeFactoX determines whether a news piece is credible or misleading and provides a well-reasoned justification, enhancing transparency and trust in automated misinformation detection.

I see several important aspects of these claims:
1. The importance of curriculum learning
2. The novelty of the framework
3. The alignment of machine-generated explanations with human reasoning
4. The factuality of the explanations

Of these claims, I believe only (4) is partially supported in the original paper, and that's if we trust the metrics to measure factuality.  A major issue is that the metrics don't clearly align with these objectives. Reference-based metrics for open-ended tasks like explanation generation have numerous issues (documented for summarization: they disagree with human evaluation on strong systems https://arxiv.org/pdf/2209.12356 ). So at face value, I don't think it's clear that these metrics support any of the conclusions.

The authors added quite a few results during the rebuttal period.

For claim (1): one extra experiment (in response to 5LoM) ablates the curriculum. The improvements are fairly minor.

For claim (3): human evaluation (in response to 5LoM) shows the overall quality (according to an average of some Likert ratings) of Hin-DPO vs. standard DPO. However, it doesn't make any claims about the alignment of these explanations with human reasoning.  The manual evaluation in response to reviewer Uuvr doesn't quite capture this either; it compares the efficacy of these two methods, but says nothing about their alignment.

For claim (4): the experiment in response to HQpS does say something about actuality score. However, in spite of this correlation all of the reviewers raise concerns about the GPT implementation of the factuality metric. I agree this is a very serious weakness.  Breaking news cannot be checked in this way. The alignment with human evaluation is nice to see, but this is more of a statement about the factual errors in this dataset. As the emphasized contribution of the work is about the framework, I think the framework is limited to cases where this approach can work well.

The reviewers and I ultimately felt that the claims were not substantiated well enough for publication at TMLR.

**Resubmission Of Major Revision:**

The authors may consider submitting a major revision at a later time.